# Limits to visual representational correspondence between convolutional neural networks and the human brain

Yaoda Xu [1✉] & Maryam Vaziri-Pashkam[2]

Convolutional neural networks (CNNs) are increasingly used to model human vision due to their high object categorization capabilities and general correspondence with human brain responses. Here we evaluate the performance of 14 different CNNs compared with human fMRI responses to natural and artificial images using representational similarity analysis. Despite the presence of some CNN-brain correspondence and CNNs' impressive ability to fully capture lower level visual representation of real-world objects, we show that CNNs do not fully capture higher level visual representations of real-world objects, nor those of artificial objects, either at lower or higher levels of visual representations. The latter is particularly critical, as the processing of both real-world and artificial visual stimuli engages the same neural circuits. We report similar results regardless of differences in CNN architecture, training, or the presence of recurrent processing. This indicates some fundamental differences exist in how the brain and CNNs represent visual information.

[1] Psychology Department, Yale University, New Haven, CT, USA. [2] Laboratory of Brain and Cognition, National Institute of Mental Health, Bethesda, MD, USA.
✉email: xucogneuro@gmail.com

Recent hierarchical convolutional neural networks (CNNs) have achieved human-like object categorization performance[1–4]. It has additionally been shown that representations formed in lower and higher layers of the network track those of the human lower and higher visual processing regions, respectively[5–8]. Similar results have also been obtained in monkey neurophysiological studies[9,10]. CNNs incorporate the known architectures of the primate lower visual processing regions and then repeat this design motif multiple times. Although the detailed neural mechanisms governing high-level primate vision remain largely unknown, the brain–CNN correspondence has generated the excitement that perhaps the algorithms governing high-level vision would automatically emerge in CNNs to provide us with a shortcut to fully understand and model primate vision. Consequently, CNNs have been regarded by some as the current best models of primate vision (e.g., [11,12]). So much so that it has recently become common practice in human functional magnetic resonance imaging (fMRI) studies to compare fMRI measures to CNN outputs (e.g., [13–15]).

Here, we reevaluate the key fMRI finding showing that representations formed in lower and higher layers of the CNN could track those of the human lower and higher visual processing regions, respectively. Our goal here is neither to deny that CNNs can capture some aspects of brain responses better than previous models nor to enter a "glass half empty" vs. "glass half full" subjective debate. But rather, we aim to evaluate CNN modeling as a viable scientific method to understand primate vision and whether there are fundamental differences in visual processing between the brain and CNNs that would limit CNN modeling as a shortcut for understanding primate vision.

Two approaches have been previously used for establishing a close brain and CNN representation correspondence[5–8]. One approach has used linear transformation to link individual fMRI voxels to the units of CNN layers through training and cross-validation[6,7]. While this is a valid approach, it is computationally costly and requires large amounts of training data to map a large number of fMRI voxels to an even larger number of CNN units. The other approach has bypassed this direct voxel-to-unit mapping, and instead, has examined the correspondence in visual representational structures between the human brain and CNNs using representational similarity analysis (RSA[16]). With this approach, both Khaligh-Razavi and Kriegeskorte[8] and Cichy et al.[5] reported a close correspondence in the representational structure of lower and higher human visual areas to lower and higher CNN layers, respectively. Khaligh-Razavi and Kriegeskorte[8] additionally showed that such correlations exceeded the noise ceiling for both brain regions, indicating that the representations formed in a CNN could fully capture those of human visual areas (but see ref. [17]).

These human findings are somewhat at odds with results from neurophysiological studies showing that the current best CNNs can only capture about 50–60% of the explainable variance of macaque V4 and IT[9,10,18,19]. Khaligh-Razavi and Kriegeskorte[8] and Cichy et al.[5] were also underpowered by a number of factors, raising concerns regarding the robustness of their findings. Most importantly, none of the above fMRI studies tested altered real-world object images (such as images that have been filtered to contain only the high or low spatial frequency components). As human participants have no trouble recognizing such filtered real-world object images, it is critical to know if a brain–CNN correspondence exists for these filtered real-world object images. Decades of vision research has successfully utilized simple and artificial visual stimuli to uncover the complexity of visual processing in the primate brain, showing that the same algorithms used in the processing of natural images would manifest themselves in the processing of artificial visual stimuli. If CNNs are to be used as working models of the primate visual brain, it is equally critical to test whether a close brain–CNN correspondence exists for the processing of artificial objects.

Here, we compared human fMRI responses from three experiments with those from 14 different CNNs (including both shallow and very deep CNNs and a recurrent CNN)[20]. In particular, following Khaligh-Razavi and Kriegeskorte[8] and Cichy et al.[5] and using the lower bound of the noise ceiling from the human brain data as our threshold, we examined how well visual representational structures in the human brain may be captured by CNNs, with "fully capture" meaning that the brain-CNN correlation would be as good as the brain-brain correlation between the human participants, which in turn would indicate that CNN is able to fully account for the total amount of explainable brain variance. We found that while a number of CNNs were successful at fully capturing the visual representational structures of lower-level human visual areas during the processing of both the original and filtered real-world object images, none could do so for these object images at higher-level visual areas. In addition, none of the CNNs tested could fully capture the visual representations of artificial objects in lower-level human visual areas, with all but one also failing to do so for these objects in higher-level human visual areas. Some fundamental differences thus exist between the human brain and CNNs and preclude CNNs from fully modeling the human visual system at their current states.

## Results

In this study, we reexamined previous findings that showed close brain–CNN correspondence in visual processing[5–8]. We noticed the two studies that used the RSA approach were underpowered in two aspects. First, both Khaligh-Razavi and Kriegeskorte[8] and Cichy et al.[5] used an event-related fMRI design, known to produce a low signal-to-noise ratio (SNR). This can be seen in the low brain–CNN correlation values reported, with the highest correlation being less than 0.2 in both studies. While Cichy et al.[5] did not calculate the noise ceiling, thus making it difficult to assess how good the correlations were, the lower bounds of the noise ceiling were around 0.15–0.2 in Khaligh-Razavi and Kriegeskorte[8], which is fairly low. Second, both studies defined human brain regions anatomically rather than functionally in each individual participant. This could affect the reliability of fMRI responses, potentially contributing to the low noise ceiling and low correlation obtained. Here, we took advantage of existing data sets from three fMRI experiments that overcome these drawbacks and compared visual processing in the human brain with those of 14 different CNNs. These data sets were collected while human participants viewed both unfiltered and filtered real-world object images and artificial object images. This allowed us to test not only the robustness of brain–CNN correlation, but also its generalization across different image sets. Because the RSA approach allows easy comparisons of multiple fMRI data sets with multiple CNNs, and because a noise ceiling can be easily derived to quantify the degree of the brain–CNN correspondence, we used this approach in the present study.

Our fMRI data were collected with a block design in which responses were averaged over a whole block of multiple exemplars to increase SNR. In three fMRI experiments, human participants viewed blocks of sequentially presented cut-out images on a gray background at fixation and pressed a response button whenever the same image repeated back to back (Fig. 1a). Each image block contained different exemplars from the same object category, with the exemplars varied in identity, viewpoint/orientation, and pose (for the animal categories) to minimize the low-level similarities among them (see Supplementary Figs. 1 and 2 for the full set of images used). A total of eight real-world natural and manmade object categories were used, including

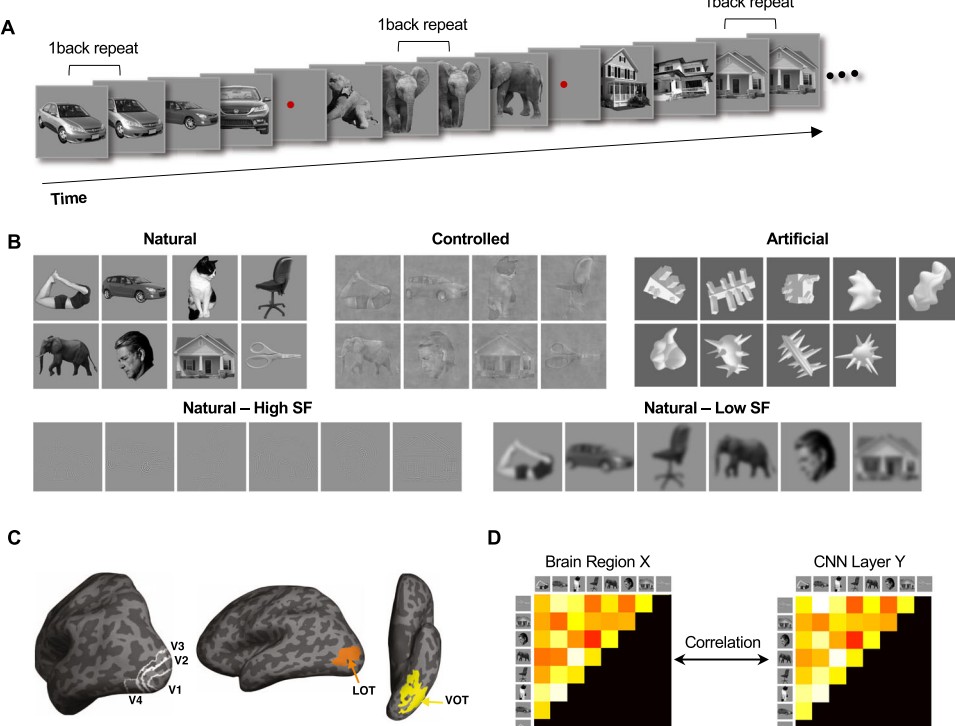

**Fig. 1 Experimental procedure and stimuli used, brain regions examined, and the representational similarity analysis method used in the study. A** An illustration of the block design paradigm used. Participants performed a one-back repetition detection task on the images. An actual block in the experiment contained ten images with two repetitions per block. See "Methods" for more details. **B** The stimuli used in the three fMRI experiments. Experiment 1 included the original and the controlled images from eight real-world object categories. Experiment 2 included the images from six of the eight real-world object categories shown in the original, high SF, and low SF format. Experiment 3 included images from the same eight real-world object categories and images from nine artificial object categories. Each category contained ten different exemplars varying in identity, viewpoint/orientation, and pose (for the animal categories) to minimize the low-level image similarities among them. See Supplementary Figs. 1 and 2 for the full set of images used. **C** The human visual regions examined. They included topographically defined early visual areas V1–V4 and functionally defined higher object processing regions LOT and VOT. **D** The representational similarity analysis used to compare the representational structural between the brain and CNNs. In this approach, a representational dissimilarity matrix was first formed by computing all pairwise Euclidean distances of fMRI response patterns or CNN layer output for all the object categories. The off-diagonal elements of this matrix were then used to form a representational dissimilarity vector. These dissimilarity vectors were correlated between each brain region and each sampled CNN layer to assess the similarity between the two. **C** is reproduced from Xu and Vaziri-Pashkam[61] with permission.

bodies, cars, cats, chairs, elephants, faces, houses, and scissors[21,22]. In Experiment 1, both the original images and the controlled version of the same images were shown (Fig. 1b). Controlled images were generated using the SHINE technique to achieve spectrum, histogram, and intensity normalization and equalization across images from the different categories[23]. In Experiment 2, the original, high and low SF contents of an image from six of the eight real-world object categories were shown (Fig. 1b). In Experiment 3, both the images from the eight real-world image categories and images from nine artificial object categories[24] were shown (Fig. 1b).

For a given brain region, we averaged fMRI responses from a block of trials containing exemplars of the same category and extracted the beta weights (from a general linear model) for the entire block from each voxel. The responses from all the voxels in a given region were then taken as the fMRI response pattern for that object category in that brain region. Following this, fMRI response patterns were extracted for each category from six independently defined visual regions along the human occipito-temporal cortex (OTC). They included lower visual areas V1 to V4 and higher visual object processing regions in lateral occipito-temporal (LOT) and ventral occipito-temporal (VOT) cortex (Fig. 1c). LOT and VOT have been considered as the homolog of the macaque inferotemporal (IT) cortex involved in visual object

processing[25]. Their responses have been shown to correlate with successful visual object detection and identification[26,27], and their lesions have been linked to visual object agnosia[28,29].

The 14 CNNs we examined here included both shallower networks, such as Alexnet, VGG16, and VGG 19, and deeper networks, such as Googlenet, Inception-v3, Resnet-50, and Resnet-101 (Supplementary Table 1). We also included a recurrent network, Cornet-S, that has been shown to capture the recurrent processing in macaque IT cortex with a shallower structure[12,19]. This CNN is argued to be the current best model of the primate ventral visual regions[19]. All CNNs were pretrained with ImageNet images[30]. To understand how the specific training images would impact CNN representations, we also examined Resnet-50 trained with stylized ImageNet images[31]. Following a previous study (O'Connor et al., 2018[32]), we sampled from 6 to 11 mostly pooling layers of each CNN (see Supplementary Table 1 for the CNN layers sampled). Pooling layers were selected because they typically mark the end of processing for a block of layers when information is pooled to be passed on to the next block of layers. We extracted the response from each sampled CNN layer for each exemplar of a category and then averaged the responses from the different exemplars to generate a category response, similar to how an fMRI category response was extracted. Following Khaligh-Razavi and Kriegeskorte[8] and

Cichy et al.[5], using RSA, we compared the representational structures of real-world and artificial object categories between the different CNN layers and different human visual regions.

**The existence of brain–CNN correspondence for representing real-world object images.** In Experiments 1 and 2, we wanted to verify the previously reported brain–CNN correspondence for representing real-world object images. We also tested if this finding can be generalized to filtered real-world images.

To compare the representational structure between the human brain and CNNs, in each brain region examined, we first calculated pairwise Euclidean distances of the z-normalized fMRI response patterns among the different object categories in each experiment, with shorter Euclidean distance indicating greater similarity between a pair of fMRI response patterns. From these pairwise Euclidean distances, we constructed a category representational dissimilarity matrix (RDM, see Fig. 1d) for each of the six brain regions examined. Likewise, from the z-normalized category responses of each sampled CNN layer, we calculated pairwise Euclidean distances among the different categories to form a CNN category RDM for that layer. We then correlated category RDMs between brain regions and CNN layers using Spearman rank correlation following Nili et al.[33] and Cichy et al.[5] (Fig. 1d). A Spearman rank correlation compares the representational geometry between the brain and a CNN without requiring the two to have a strictly linear relationship. All our results remained the same when Pearson correlation was applied and when correlation measures, instead of Euclidean distance measures, were used to construct the category RDMs (see Supplementary Figs. 3, 6, 7, and 16).

Previous studies have reported a correspondence in representation between lower and higher CNN layers to lower and higher visual processing regions, respectively[5,8]. To evaluate the presence of such correspondence in our data, for each CNN, we identified the layer that showed the best RDM correlation with each of the six included brain regions in each participant. We then assessed whether the resulting layer numbers increased from low-to-high visual regions using Spearman rank correlation. If a close brain–CNN correspondence in representation exists, then the Fisher-transformed correlation coefficient of this Spearman rank correlation should be significantly above zero at the group level (one-tailed $t$ tests were conducted to test for significance; one-tailed $t$ tests were used as only values above zero are meaningful; all stats reported were corrected for multiple comparisons for the number of comparisons included in each experiment using the Benjamini–Hochberg procedure at false discovery rate $q = 0.05$, see ref. [34]).

In Experiment 1, we contrasted original real-world object images with the controlled version of these images. Figure 2a shows the average CNN layer that best correlated with each brain region for each CNN during the processing of these images (the exact significance levels of the brain–CNN correspondence are marked with asterisks at the top of each plot). Here, 10 out of the 14 CNNs examined showed a significant brain–CNN correspondence for the original images. The same correspondence was also seen for the controlled images, with 11 out of the 14 CNNs showing a significant brain–CNN correspondence.

In Experiment 2, we contrasted original real-world images with the high and low SF component versions of these images (Fig. 2b). For the original images, we replicated the findings from Experiment 1, with 13 out of the 14 CNNs showing a significant brain–CNN correspondence. The same correspondence was also present in 13 CNNs for the high SF images and in 8 CNNs for the low SF images. In fact, Alexnet, Cornet-S, Googlenet, Inception-v3, Mobilenet-v2, Resnet-18, Resnet-50, Squeezenet, and VGG16 showed a significant

brain–CNN correspondence for all five image sets across the two experiments. These results remained the same when correlations, instead of Euclidean distance measures, were used to construct the category RDMs, and Pearson, instead of Spearman, the correlation was applied to compare CNN and brain RDMs (Supplementary Fig. 3).

These results replicate previous findings using the RSA approach[5,8] and show that there indeed existed a brain–CNN correspondence, with representations in lower and higher visual areas better resembling those of lower and higher CNN layers, respectively. Importantly, such a brain–CNN correspondence is generalizable to filtered real-world object images.

**Quantifying the amount of brain–CNN correspondence for representing real-world object images.** A linear correspondence between CNN and brain representations, however, only tells us that lower CNN layers are relatively more similar to lower than higher visual areas and that the reverse is true for higher CNN layers. It does not tell us about the amount of similarity. To assess this, we evaluated how successfully the category RDM from a CNN layer could capture the RDMs from a brain region. To do so, we first obtained the reliability of the category RDM in a brain region across human participants by calculating the lower and upper bounds of the fMRI noise ceiling[33]. Overall, the lower bounds of fMRI noise ceilings for the different brain regions were much higher in our two experiments than those of Khaligh-Razavi and Kriegeskorte[8] (Supplementary Figs. 4A and 5A). These results indicate that the object category representational structures in our data are fairly similar and consistent across participants.

If the category RDM from a CNN layer successfully captures that from a brain region, then the correlation between the two should exceed the lower bound of the fMRI noise ceiling. This can be re-represented as the proportion of explainable brain RDM variance captured by the CNN (by dividing the brain–CNN RDM correlation by the lower bound of the corresponding noise ceiling and then taking the square of the resulting ratio; all correlation results are reported in Supplementary Figs. 4–7). For the original real-world object images in Experiment 1, the brain RDM variance from lower visual areas was fully captured by three CNNs (Fig. 3a), including Alexnet, Googlenet, and Vgg16 (with no difference between 1 and the highest proportion of variance explained by a CNN layer for V1–V3, one-tailed $t$ tests, $ps > 0.1$; see the asterisks marking the exact significance levels at the top of each plot; one-tailed $t$ tests were used here as only testing the values below 1 was meaningful; all $p$ values reported were corrected for multiple comparisons for the 6 brain regions included using the Benjamini–Hochberg procedure at false discovery rate $q = 0.05$). However, no CNN layer was able to fully capture the RDM variance from visual areas LOT and VOT (with significant differences between 1 and the highest proportion of variance explained by a CNN layer for LOT and VOT, $ps < 0.05$, one-tailed and corrected). The same pattern of results was observed when the controlled images were used in Experiment 1 (Fig. 3b): several CNNs were able to fully capture the RDM variance of lower visual areas but none was able to do so for higher visual areas. We obtained similar results for the original, high SF, and low SF images in Experiment 2 (Fig. 4a–c). Here again, a number of CNNs fully captured the RDM variance of lower visual areas, but none could do so for higher visual areas. All these results remained the same when correlations, instead of Euclidean distance measures, were used to construct the category RDMs, and Pearson, instead of Spearman, correlations were applied to compare CNN and brain RDMs (see the correlation results in Supplementary Figs. 4–7; note that although using

**A    Experiment 1**

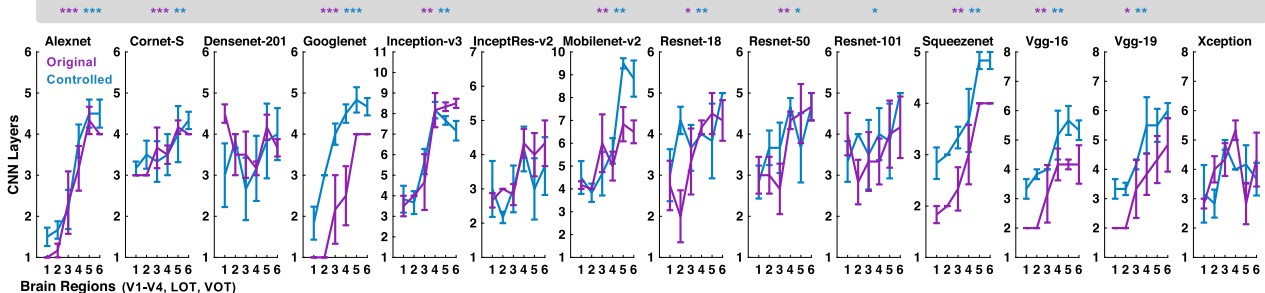

**B    Experiment 2**

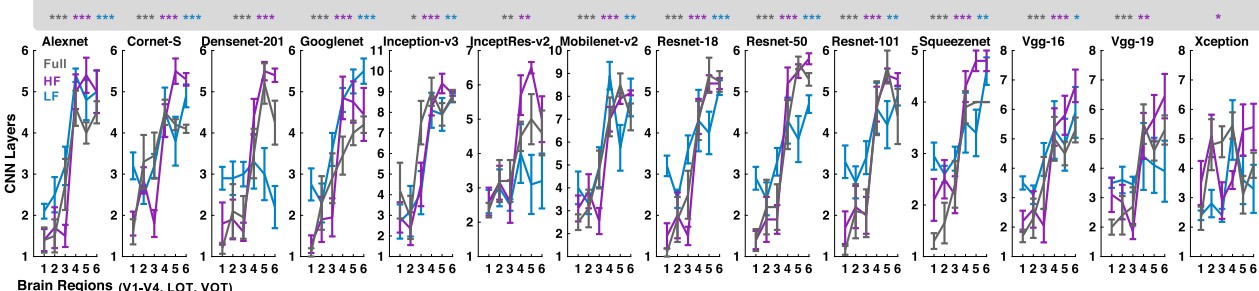

**C    Experiment 3**

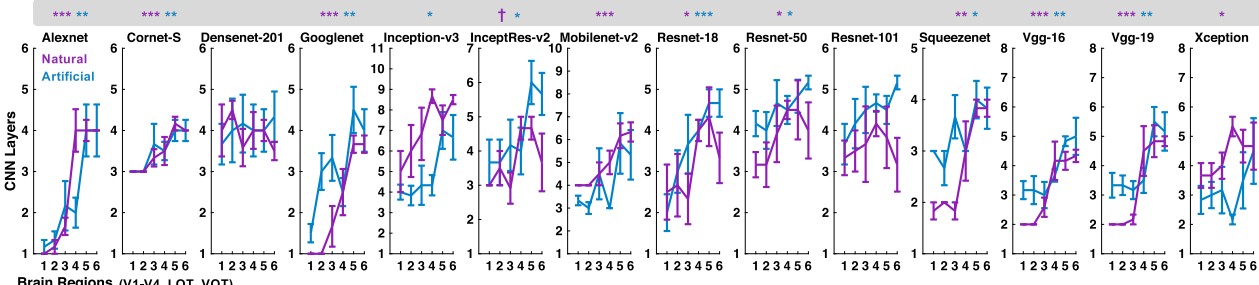

**Fig. 2 Evaluation of brain–CNN correspondence in representational structures. A** The results from Experiment 1, in which original and controlled images from real-world object categories were shown. $N = 6$ human participants. **B** The results from Experiment 2, in which full, high and low SF components of the images from real-world object categories were shown. $N = 10$ human participants. **C** The results from Experiment 3, in which unaltered images from both real-world (natural) and artificial object categories were shown. $N = 6$ human participants. Plotting here are the averaged CNN layer numbers across the human participants that showed the greatest RDM correlation for each brain region in each experimental condition, with the error bars indicating the standard errors of the mean across participants. To evaluate brain–CNN correspondence, in each human participant, the CNN layer that showed the highest RDM correlation with each of the six brain regions was identified. A Spearman rank correlation was carried out for each participant to assess whether the resulting layer numbers increased from low to high human visual regions. The resulting correlation coefficients (Fisher-transformed) were tested for greater than zero at the participant group level using one-tailed $t$ tests. The asterisks at the top of each plot mark the significance level of these statistical tests, with a significant result indicating that the RDMs from lower CNN layers better correlated with those of lower than higher visual regions and the reverse is true for higher CNN layers. All $t$-tests were corrected for multiple comparisons for the number of image conditions included in each experiment using the Benjamini–Hochberg procedure. $^{\dagger}p < 0.1$, $^*p < 0.05$, $^{**}p < 0.01$, $^{***}p < 0.001$. Source data are provided as a Source Data file.

Euclidean distance measures after pattern z-normalization to construct the RDMs produced highly similar results as those from correlation measures, they were not identical).

In our fMRI experiments, we used a randomized presentation order for each of the experimental runs with two image repetitions. When we simulated the exact fMRI design in Alexnet by generating a matching number of randomized presentation sequences with image repetitions and then averaging CNN responses for these sequences, we obtained virtually identical Alexnet results as those without this simulation (Supplementary Fig. 4D). Thus, the disagreement between our fMRI and CNN

results could not be due to a difference in stimulus presentation. The very fact that CNN could fully capture the brain RDM variance in lower visual areas for real-world objects further supports this idea and additionally shows that the non-linearity in fMRI measures had a minimal impact on RDM extraction. The latter speaks to the robustness of the RSA approach as extensively reviewed elsewhere[16].

Together, these results showed that, although lower layers of several CNNs could fully capture the explainable brain RDM variance for lower-level visual representations of both the original and filtered real-world object images in the human brain, none

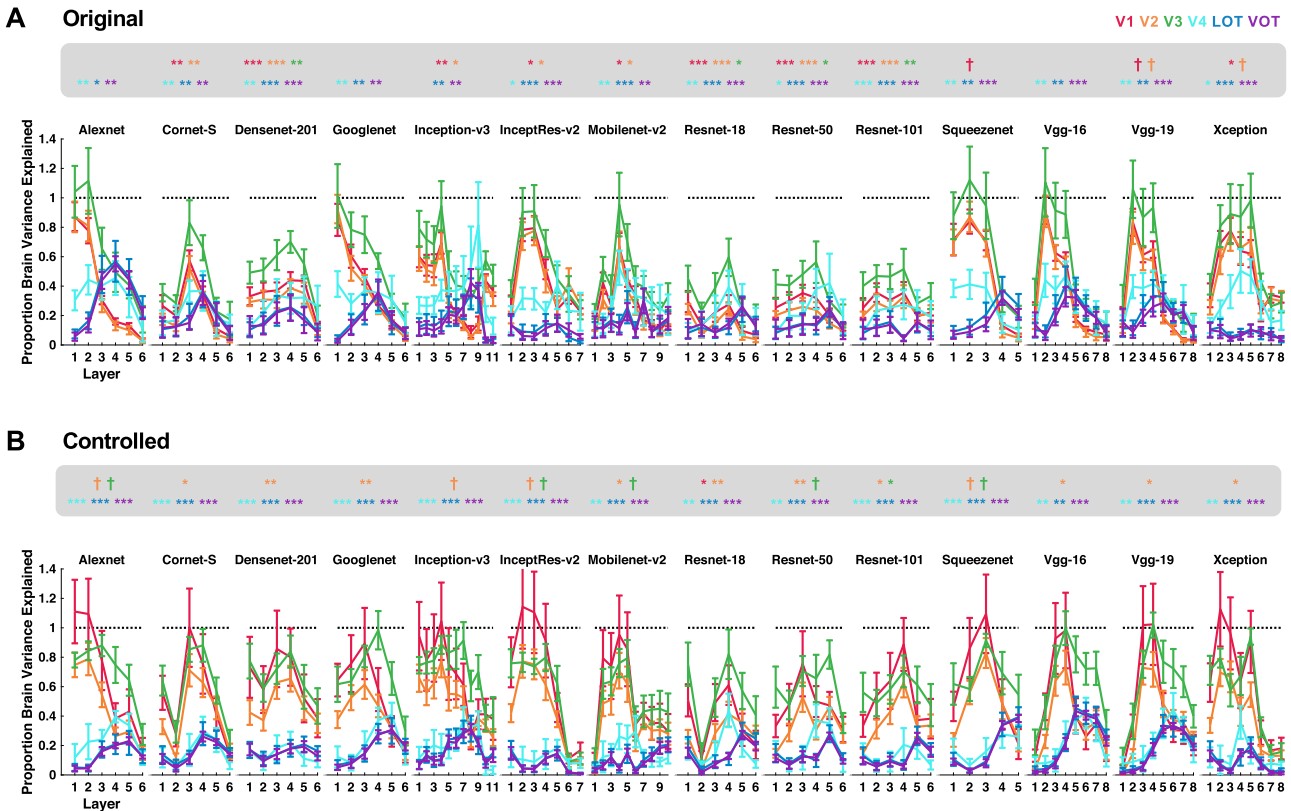

**Fig. 3 Quantifying the proportion of explainable brain RDM variance captured by each sampled CNN layer in Experiment 1 for the processing of images from real-world object categories. A** Results for the Original images. **B** Results for the Controlled images. $N = 6$ human participants. The asterisks at the top of each plot mark the significance level of the difference between 1 and the highest proportion of variance explained by a CNN for each brain region; one-tailed $t$-tests were used as only values below 1 were meaningful here; all $p$ values reported were corrected for multiple comparisons for the six brain regions included using the Benjamini–Hochberg procedure. Error bars indicate standard errors of the means. $^{†}p < 0.1$, $^{*}p < 0.05$, $^{**}p < 0.01$, $^{***}p < 0.001$. Source data are provided as a Source Data file.

could do so for higher-level neural representations of these images. In fact, the highest amount of explainable brain RDM variance that could be captured by CNNs from higher visual regions LOT and VOT was about 60%, on par with previous neurophysiological results from macaque IT cortex[9,10,18,19].

To directly visualize the object representational structures in different brain regions and CNN layers, using multi-dimensional scaling (MDS, Shepard, 1980[35]), we placed the RDMs on 2D spaces with the distances among the categories approximating their relative similarities to each other. Figure 5a, b shows the MDS plots from the two lowest and the two highest brain regions examined (i.e., V1, V2, LOT, and VOT) and from the two lowest and the two highest layers sampled from four examples CNNs (i.e., Alexnet, Cornet-S, Googlenet, and Vgg-19) from Experiments 1 and 2 (see Supplementary Figs. 8–12 for the MDS plots from all brain regions and CNN layers sampled). Consistent with our quantitative analysis, for the real-world objects, there were some striking brain–CNN representational similarities at lower levels of object representation (such as in Alexnet and Googlenet). At higher levels, both the brain and CNNs showed a broad distinction between animate and inanimate objects (i.e., bodies, cats, elephants, and faces vs. cars, chairs, houses, and scissors), but they differed in how these categories were represented relative to each other. For example, within the animate objects, while faces and bodies are far apart in both VOT and LOT, they are next to each other in higher CNN layers (see the objects marked by the dotted circles in Fig. 5); and within the inanimate objects, while cars, chairs, houses, and scissors tend to form a square in

VOT and LOT, they tend to form a line in higher CNN layers (see the objects marked by the dashed ovals in Fig. 5).

LOT and VOT included a large swath of the ventral and lateral OTC and likely overlapped to a great extent with regions selective for specific object categories, such as faces, bodies, or scenes. Because CNNs may not automatically develop category-selective units during object categorization training, it is possible that the brain–CNN RDM discrepancy we observed so far at higher levels of visual processing is solely driven by the category-selective voxels in the human brain. To investigate this possibility, using the main experimental data, we evaluated the category selectivity of each voxel in LOT and VOT (see "Methods"). We then excluded all voxels showing a significant category selectivity for faces, bodies, or scenes (i.e., houses) and repeated our analysis. In most cases, the amount of the brain RDM variance that could be capture by CNNs remained unchanged whether or not category-selective voxels were included or excluded (see Supplementary Figs. 13 and 14). Significant differences were observed in only 6% of the comparisons ($ps < 0.05$, uncorrected, see the caption of Supplementary Figs. 13 and 14 for a list of these cases). However, even in these cases, the maximum amount of LOT and VOT RDM variance captured by CNNs was still significantly less than 1 ($ps < 0.05$, corrected). Moreover, when the same unaltered images were shown across the different experiments, the improvement seen in one experiment was not replicated in another experiment (e.g., the improvement seen in Alexnet for Experiment 2 Full-SF was not replicated in Experiment 3 Natural, see Supplementary Figs. 14 and 18). Consistent with these results,

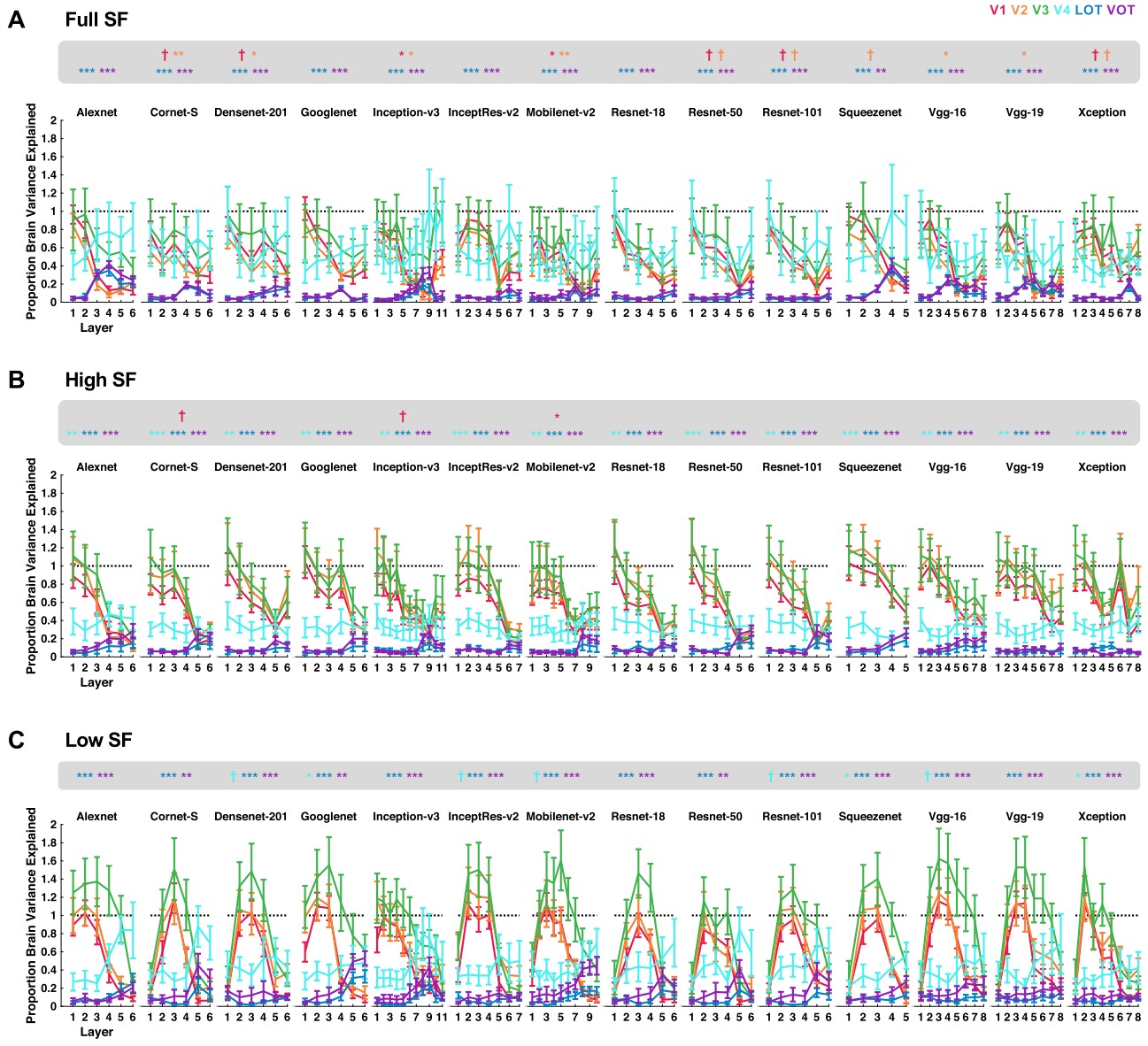

**Fig. 4 Quantifying the proportion of explainable brain RDM variance captured by each sampled CNN layer in Experiment 2 for the processing of images from real-world object categories. A** Results for Full SF images. **B** Results for High SF images. **C** Results for Low SF images. $N = 10$ human participants. The asterisks at the top of each plot mark the significance level of the difference between 1 and the highest proportion of variance explained by a CNN for each brain region; one-tailed $t$ tests were used and all $p$ values reported were corrected for multiple comparisons for the six brain regions included using the Benjamini–Hochberg procedure. Error bars indicate standard errors of the means. †$p < 0.1$, *$p < 0.05$, **$p < 0.01$, ***$p < 0.001$. Source data are provided as a Source Data file.

MDS plots for LOT and VOT look quite similar whether or not category-selective voxels were included (see Supplementary Figs. 8–12). As such, the failure of CNNs to fully capture brain RDM at higher levels of visual processing cannot be attributed to the presence of category-selective voxels in LOT and VOT.

One could argue that CNNs generally do not encounter disembodied heads or headless bodies in their training data. They are thus unlikely to have distinctive representations for heads and bodies. Note that the human visual system generally does not see such stimuli in its training data either. The goal of the study is, therefore, not to test images that a system has been exposed to during training, but rather how it handles images that it has not. If the two systems are similar in their underlying representation, then they should still respond similarly to images that they have not been exposed to during training. If not, then it indicates that

the two systems represent visual objects in different ways. We present a stronger test case in the next experiment by comparing the representations of artificial visual stimuli between the brain and CNNs.

**The brain–CNN correspondence for representing artificial object images.** Previous comparisons of visual processing in the brain and CNN have focused entirely on the representation of real-world objects. Decades of visual neuroscience research, however, has successfully utilized simple and artificial visual stimuli to uncover the complexity of visual processing in the primate brain (e.g., [36–39]), with Tanaka and colleagues, in particular, showing that IT responses to some real-world objects are highly similar to their responses to artificial shapes[39]. The same algorithms used in the processing of natural images thus manifest

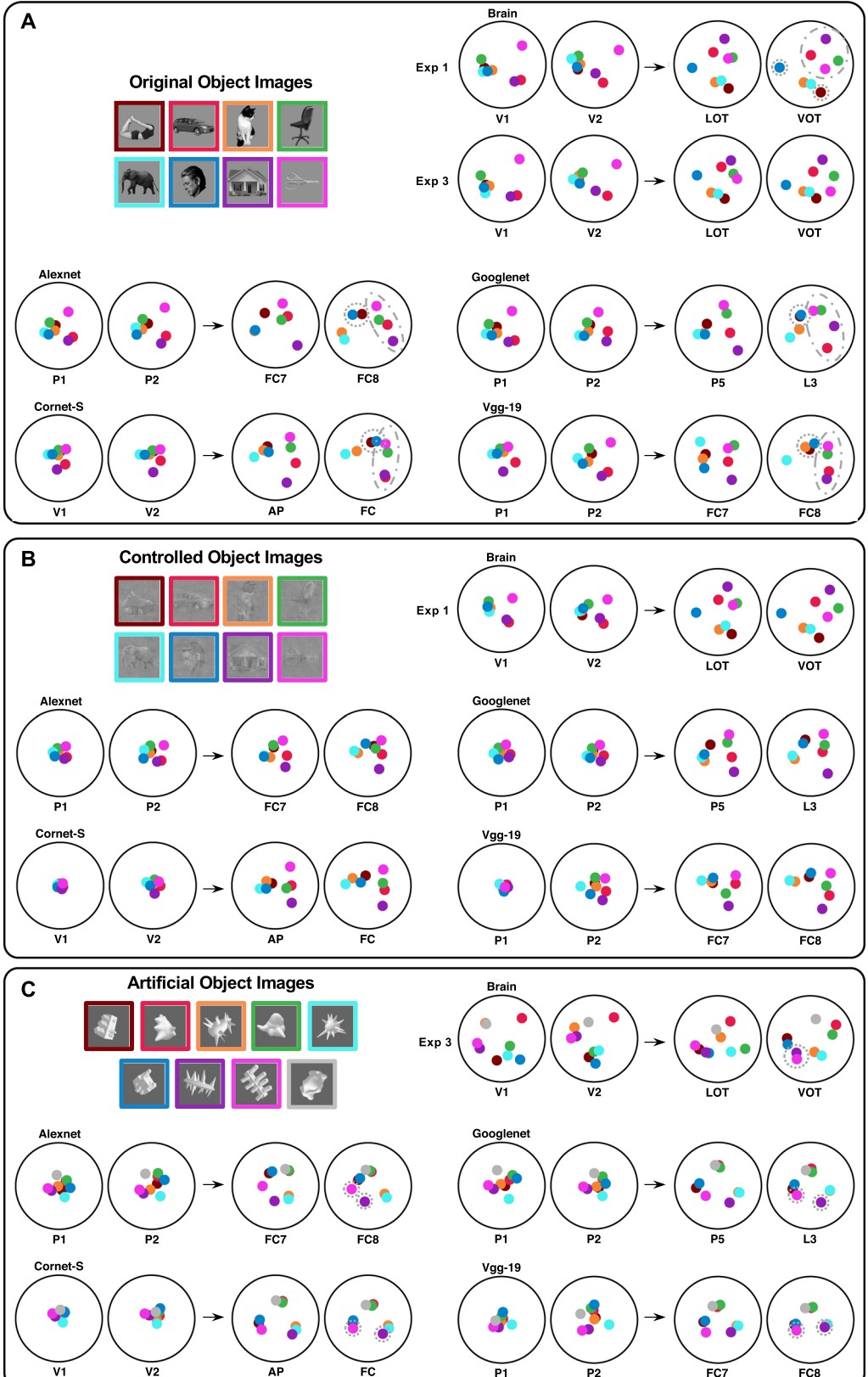

themselves in the processing of artificial visual stimuli. If CNNs are to be used as working models of the primate visual brain, it would be critical to test if this principle applies to CNNs.

Testing simple and artificial visual stimuli also allows us to address a remaining concern for the results obtained so far. It could be argued that the reason CNNs performed poorly in fully

tracking high-level processing of the real-world objects even when category-selective voxels were removed was due to interactions between category-selective and non-selective brain regions. With the artificial visual stimuli, however, no preexisting category information, semantic knowledge, as well as experience with the stimuli could affect visual processing at a higher level. This would

**Fig. 5 Visualizing the object representational structure in human visual regions and CNN layers using MDS. A** Results for the Original real-world object images. **B** Results for the Controlled real-world object images. **C** Results for the artificial object images. Brain responses included here are those for the original real-world images from both Experiments 1 and 3, those for the controlled real-world images from Experiment 1, and those for the artificial object images from Experiment 3. The distances among the object categories in each MDS plot approximate their relative similarities to each other in the corresponding RDM. Only MDS plots from the two lowest and the two highest brain regions examined (i.e., V1, V2, LOT, and VOT) and from the two lowest and two highest layers sampled from four examples, CNNs (i.e., Alexnet, Cornet-S, Googlenet, and Vgg-19) are included here. See Supplementary Figs. 8–12 and 17 for MDS plots from all brain regions and CNN layers examined. Since rotations and flips preserve distances on these MDS plots, to make these plots more informative and to see how the representational structure evolved across brain regions and CNN layers, we manually rotated and/or flipped each MDS when necessary. For real-world objects, there were some remarkable brain–CNN similarities at lower levels of object representations (see Alexnet and Googlenet). At higher levels, although both showed a broad distinction between animate and inanimate objects (i.e., bodies, cats, elephants, and faces vs. cars, chairs, houses, and scissors), they differ in how categories are represented from each other. For example, within the animate objects, while faces and bodies are far apart in both VOT and LOT, they are next to each other in higher CNN layers (see the objects marked by the dotted circles in (**A**); and within the inanimate objects, while cars, chairs, houses, and scissors tend to form a square in VOT and LOT, they tend to form a line in higher CNN layers (see the objects marked by the dashed ovals in (**A**). For the artificial object images, brain–CNN differences at the lower level are not easily interpretable. Differences at the higher level suggest that while the brain takes both local and global shape similarities into account when grouping objects, CNNs rely mainly on local shape similarities. This can be seen in the grouping of the objects at higher CNN layers and by comparing the purple and fuchsia shapes that share the same global but different local features (see the objects marked by the dotted circles in (**C**)). Source data are provided as a Source Data file.

put the brain and CNN on even grounds. If CNNs still fail to track the processing of the artificial visual stimuli at higher levels, it would indicate some fundamental differences in how the brain and CNNs process visual information, rather than the particularity of the stimuli used.

In Experiment 3, we compared the processing of both real-world objects and artificial objects between the brain and CNNs. As in Experiments 1 and 2, the processing of real-world objects showed a consistent brain–CNN correspondence in 8 out of the 14 CNNs tested (Fig. 2c). The same correspondence was also obtained in eight CNNs when artificial objects were shown, with lower visual representations in the brain better resembling those of lower than higher CNN layers and the reverse is true for higher visual representations in the brain (Fig. 2c and Supplementary Fig. 3). In fact, across Experiments 1–3, Alexnet, Cornet-S, Googlenet, Resnet-18, Resnet-50, Squeezenet, and VGG16 were the seven CNNs showing a consistent brain–CNN correspondence across all our image sets, including the original and filtered real-world object images, as well as the artificial object images.

As before, for real-world objects, while some of the CNNs were able to fully capture the brain RDM variance from lower visual areas, none could do so for higher visual areas (Fig. 6a). For artificial object images, while the majority of the CNNs still failed to fully capture the brain RDM variance of higher visual areas, surprisingly, no CNN was able to do so for lower visual areas anymore (with significant differences between 1 and the highest proportion of variance explained by a CNN layer for V1 and V2, all $ps < 0.05$, one-tailed and corrected; see the asterisks marking the exact significance levels at the top of each plot for the full stats). In fact, the amount of the brain RDM variance captured in lower visual areas dropped significantly or marginally significantly between the natural and artificial objects in several CNNs (Alexnet, $p = 0.062$ for V1, $p = 0.074$ for V2; Googlenet, $p = 0.012$ for V1, $p = 0.023$ for V2; Mobilenet-v2, $p = 0.032$ for V2; Squeezenet, $p = 0.022$ for V1, $p = 0.0085$ for V2; Vgg-16, $p = 0.003$ for V1, $p = 0.0042$ for V2, $p = 0.094$ for V3; and Vgg-19, $p = 0.048$ for V1, $p = 0.0077$ for V2; all reported $p$ values were corrected for multiple comparisons for the six brain regions examined). This rendered the few CNNs that were capable of fully capturing the brain variance from the lower visual areas during the processing of real-world objects no longer able to do so during the processing of artificial objects (Fig. 6b; see also the correlation results in Supplementary Figs. 15 and 16). In other words, as a whole, CNNs performed much worse in capturing visual processing of artificial than real-world objects in the

human brain, and their ability to capture lower-level visual processing of real-world objects in the brain did not generalize to the processing of artificial objects.

For artificial objects, RDM differences between lower brain regions and lower CNN layers were not easily interpretable from the MDS plots (Fig. 5c and Supplementary Fig. 17). RDM differences between higher brain regions and higher CNN layers suggest that while the brain takes both local and global shape similarities into consideration when grouping objects, CNNs rely mainly on local shape similarities (e.g., compare higher brain and CNN representations of the shapes marked by purple and fuchsia colors that share the same global but different local features; see the objects marked by the dotted circles in Fig. 5c). This is consistent with other findings that specifically manipulated local and global shape similarities (see "Discussion"). Lastly, as in Experiments 1 and 2, removing the category-selective voxels in LOT and VOT did not improve CNN performance (see Supplementary Fig. 18).

Overall, taking both the linear correspondence and RDM correlation into account, none of the CNNs examined here could fully capture lower or higher levels of neural processing of artificial objects. This is particularly critical given that a number of CNNs were able to fully capture the lower-level neural processing of real-world objects.

**The effect of training a CNN on original vs. stylized image-net images.** Although CNNs are believed to explicitly represent object shapes in the higher layers[1,40,41], emerging evidence suggests that CNNs may largely use local texture patches to achieve successful object classification[42,43] or local rather than global shape contours for object recognition[44]. In a recent demonstration, CNNs were found to be poor at classifying objects defined by silhouettes and edges. In addition, when texture and shape cues were in conflict, they classified objects according to texture rather than shape cues[31] (see also ref. [44]). However, when Resnet-50 was trained with stylized ImageNet images in which the original texture of every single image was replaced with the style of a randomly chosen painting, object classification performance significantly improved, relied more on shape than texture cues, and became more robust to noise and image distortions[31]. It thus appears that a suitable training data set may overcome the texture bias in standard CNNs and allow them to utilize more shape cues.

We tested if the category RDM in a CNN may become more brain-like when a CNN was trained with stylized ImageNet images. To do so, we compared the representations formed in

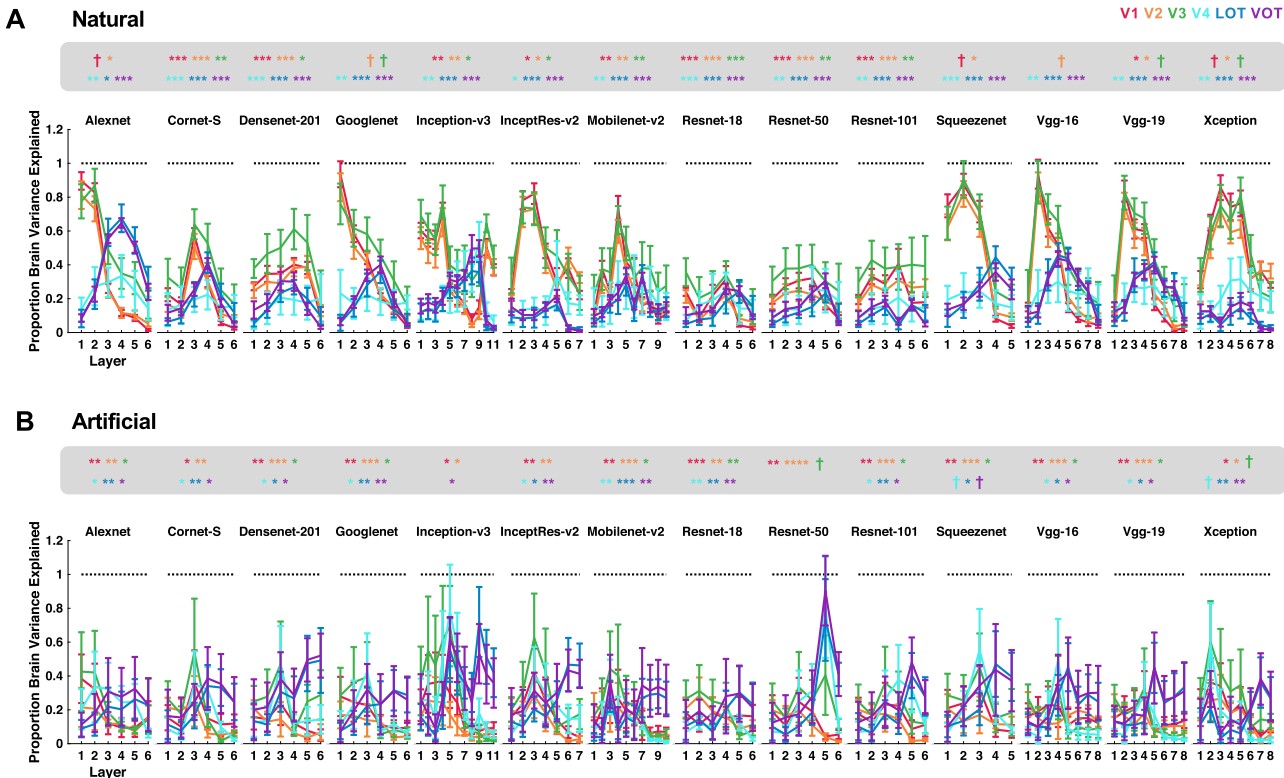

**Fig. 6 Quantifying the proportion of explainable brain RDM variance captured by each sampled CNN layer in Experiment 3. A** Results for real-world object images. **B** Results for artificial object images. $N = 6$ human participants. The asterisks at the top of each plot mark the significance level of the difference between 1 and the highest proportion of variance explained by a CNN for each brain region; one-tailed $t$ tests were used and all $p$ values reported were corrected for multiple comparisons for the six brain regions included using the Benjamini–Hochberg procedure. Error bars indicate standard errors of the means. $^{\dagger}p < 0.1$, $^{*}p < 0.05$, $^{**}p < 0.01$, $^{***}p < 0.001$. Source data are provided as a Source Data file.

Resnet-50 pretrained with ImageNet images with those from Resnet-50 pretrained with three other procedures:[31] trained only with the stylized ImageNet Images, trained with both the original and the stylized ImageNet Images, and trained with both sets of images and then fine-tuned with the stylized ImageNet images. Despite differences in training, the category RDM correlations between brain regions and CNN layers were remarkably similar among these Resnet-50s, and all were substantially different from those of the human visual regions (Supplementary Fig. 19). If anything, training with the original ImageNet images resulted in a better brain–CNN correspondence in several cases than the other training conditions. The incorporation of stylized ImageNet images in training thus did not result in more brain-like visual representations in Resnet-50.

## Discussion

It has become common practice in recent human fMRI research to regard CNNs as a working model of the human visual system. This is largely based on fMRI studies showing that representations formed in CNN lower and higher layers track those of the human lower and higher visual processing regions, respectively[5–8]. Here, we reevaluated this finding with more robust fMRI data sets from 3 experiments and 14 different CNNs and tested the generality of this finding to filtered real-world object images and artificial object images.

We found a significant correspondence in visual representational structure between the CNNs and the human brain across various image manipulations for both real-world and artificial object images, with representations formed in CNN lower layers more closely resembling those of lower than higher human visual areas and the reverse being true for higher CNN layers. In

addition, we found that lower layers of several CNNs fully captured the representational structures of real-world objects of human lower visual areas for both the original and the filtered versions of these images. This replicated earlier results and showed that CNNs are capable of capturing some aspects of visual processing in the human brain.

Despite these successes, however, no CNN tested could fully capture the representational structure of the real-world object images in human higher visual areas. The same results were obtained regardless of whether or not category-selective voxels were included in human higher visual areas. Overall, the highest amount of explainable brain RDM variance that could be captured by CNNs from higher visual regions was about 60%. This is in agreement with previous neurophysiological studies on Macaque IT cortex[9,10,18,19]. When artificial object images were used, not only did most of the CNNs still fail to capture visual processing in higher human visual areas but also none could do so for lower human visual areas. Overall, no CNN examined could fully capture all levels of visual processing for both real-world and artificial objects, with similar performance observed in both shallow and deep CNNs (e.g., Alexnet vs. Googlenet). Although the recurrent CNN examined here, Cornet-S closely models neural processing and is argued to be the current best model of the primate ventral visual regions[12,19], it did not outperform the other CNNs. The same results were also obtained when a CNN was trained with stylized object images that emphasized shape features in its representation. The main results across the three experiments are summarized in Fig. 7, with Fig. 7a showing the results from the six conditions across the three experiments examining the real-world objects (i.e., the results from Figs. 3, 4, and 6a) and Fig. 7b showing the results for

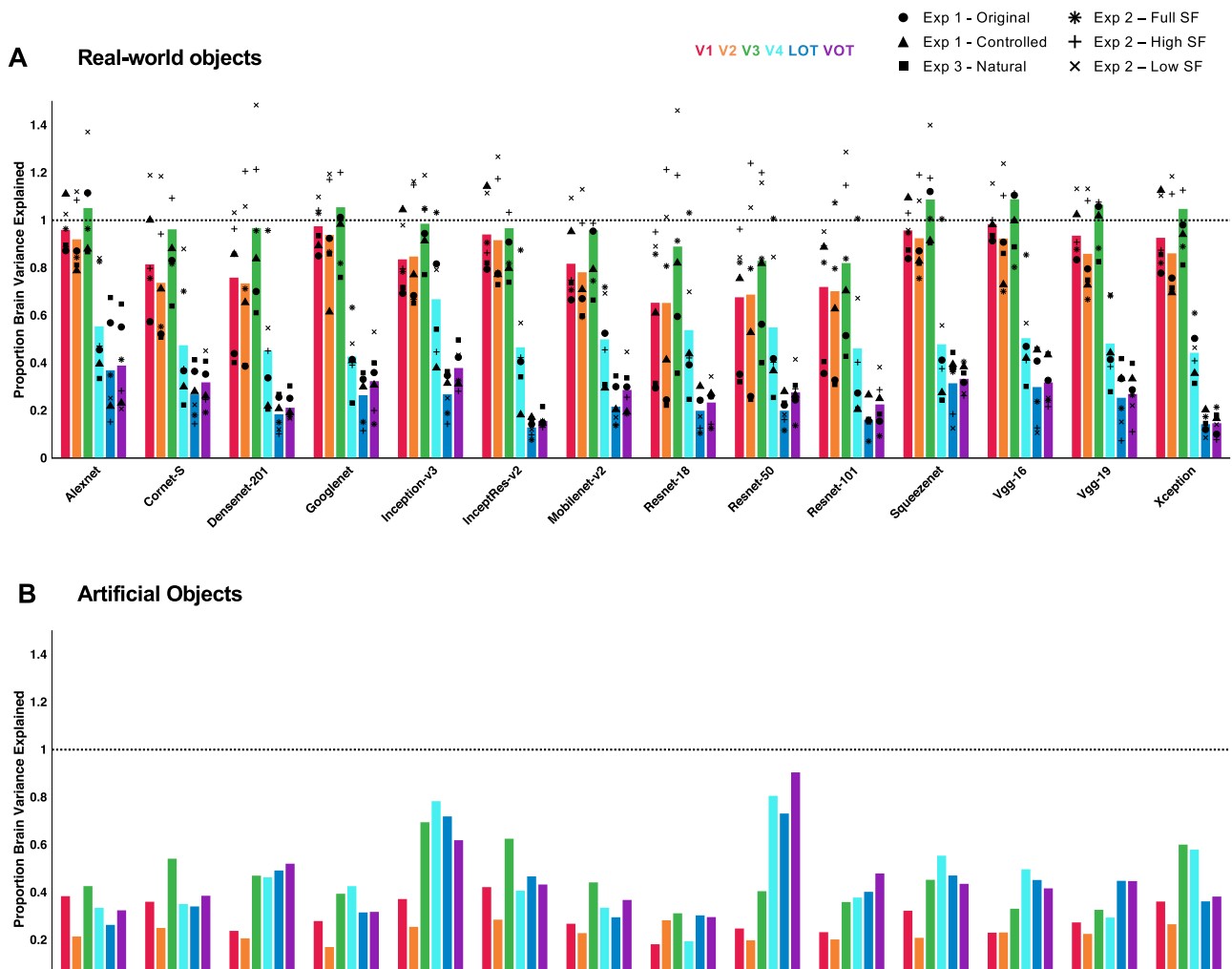

**Fig. 7 Summary of results from all three experiments. A** Summary of results from the six conditions across the three experiments that examined the processing of real-world object images (i.e., a summary of results from Figs. 3, 4, and 6). **B** Summary of results for the processing of artificial objects (i.e., results from Fig. 6b). In **A**, each colored bar represents the averaged proportion of brain variance explained, with that from each condition marked by a black symbol. For real-world objects, a few CNNs (i.e., Alexnet, Googlenet, Squeezenet, and Vgg-16) were able to consistently capture brain RDM variance from lower human visual regions (i.e., V1–V3). No CNN was able to do so for higher human visual regions (i.e., LOT and VOT). The CNNs capable of fully capturing lower-level brain RDM variance for real-world objects all failed to capture that of the artificial objects from neither lower nor higher human visual regions. Source data are provided as a Source Data file.

the artificial objects (i.e., the results from Fig. 6b). Alexnet, Googlenet, Squeezenet, and Vgg-16 showed the best brain–CNN correspondence overall for representing real-world objects among the 14 CNNs examined.

Although we examined object category responses averaged over multiple exemplars rather than responses to each object, previous research has shown similar category and exemplar response profiles in macaque IT and human lateral occipital cortex with more robust responses for categories than individual exemplars due to an increase in SNR[45,46]. Rajalingham et al.[2] additionally reported better behavior-CNN correspondence at the category but not at the individual exemplar level. Thus, comparing the representational structure at the category level, rather than at the exemplar level, should have increased our chance of finding a close brain–CNN correspondence. Yet despite the overall brain and CNN correlations for object categories being much higher here than in previous studies for individual

objects[5,8], CNNs failed to fully capture the representational structure of real-world objects in the human brain and performed even worse for artificial objects. Object category information is shown to be better represented by higher than lower visual regions (e.g., [47]). Our use of object category was thus not optimal for finding a close brain–CNN correspondence at lower levels of visual processing. Yet we found better brain–CNN correspondence at lower than higher levels of visual processing for real-world object categories. This suggests that information that defines the different real-world object categories is present at lower levels of visual processing and is captured by both lower visual regions and lower CNN layers. This is not surprising as many categories may be differentiated based on low-level features even with a viewpoint/orientation change, such as curvature and the presence of unique features (e.g., the large round outline of a face/head, the protrusion of the limbs in animals)[48]. Finally, it could be argued that the dissimilarity between the brain and

CNNs at higher levels of visual processing for real-world object categories could be driven by feedback from high-level nonvisual regions and/or feedback from category-selective regions in the human ventral cortex for some of the categories used (i.e., faces, bodies, and houses). However, such feedback should greatly decrease for artificial object categories. Yet we failed to see much improvement in brain–CNN correspondence at higher levels of processing for these objects. If anything, even the strong correlation at lower levels of visual processing for real-world objects no longer existed for these artificial objects.

Decades of vision science research has relied on using simple and artificial visual stimuli to uncover the complexity of visual processing in the primate brain, showing that the same algorithms used in the processing of natural images would manifest themselves in the processing of artificial visual stimuli. The artificial object images tested here have been used in previous fMRI studies to understand object processing in the human brain (e.g., Op de Beeck et al., 2008[21,24,27]). In particular, we showed that the transformation of visual representational structures across occipito-temporal and posterior parietal cortices follows a similar pattern for both the real-world objects and the artificial objects used here[21]. The disconnection between the representation of real-world and artificial object images in CNNs is in disagreement with this long-held principle in primate vision research and suggests that, even at lower levels of visual processing, CNNs differ from the primate brain in fundamental ways. Such a divergence will undoubtedly contribute to even greater divergence at higher levels of processing between the primate brain and CNNs.

Using real-world object images, recent studies have tried to improve brain and CNN RDM correlation by incorporating brain responses during CNN training. Using a recurrent network architecture, Kietzmann et al.[49] used both brain RDM and object categorization to guide CNN training and found that brain and CNN RDM correlation was still significantly below the noise ceiling in all human ventral visual regions examined. Khaligh-Razavi et al.[50] used a mixed RSA approach by first finding the best linear transformation between fMRI voxels and CNN layer units and then performing RDM correlations (see also ref. [10]). The key idea here is that CNNs may contain all the right brain features in visual processing but that these features are improperly combined. Training enables remixing and recombination of these features and can result in a better brain–CNN alignment in representational structure. Using the mixed RSA approach, Khaligh-Razavi et al.[50] reported that the correlation between brain and CNN was able to reach the noise ceiling for LO. However, brain–CNN correlations were fairly low for all brain regions examined (i.e., V1–V4 and LO), with noise ceiling being just below 0.5 in V1 to just below 0.2 in LO (thus the amount of explainable variance was less than 4% in LO, which is really low). The low LO noise ceiling again raises concerns about the robustness of this finding (as it did for ref. [8]). Khaligh-Razavi et al.[50] used a large data set from Kay et al.[51], which contained 1750 unique training images with each shown twice, and 120 unique testing images with each shown 13 times. Our data in comparison are limited, containing between 16 and 18 different stimulus conditions, each shown between 16 to 18 times. We are thus underpowered to perform the mixed RSA analysis here to provide an objective evaluation of this approach. It should be noted that applying the mixed RSA analysis is not as straightforward as it seems, as we do not fully understand the balance between decreased model performance due to overfitting and increased model performance due to feature mixing, as well as the minimum amount of data needed for training and testing. In addition, a mixed RSA approach requires brain responses from a large number of single images. This will necessarily result in lower

power and lower reliability across participants. In other words, due to noise, only a small amount of consistent neural responses are preserved across participants (as in Khaligh-Razavi et al.[50]), resulting in much of the neural data used to train the model likely just being subject-specific noise. This can significantly weaken the mixed RSA approach. In addition, whether a mixed RSA model trained with one kind of object image (e.g., real-world object images) may accurately predict the responses from another kind of object image (e.g., artificial object images) has not been tested. Thus, although the general principle of a mixed RSA approach is promising, what it can actually deliver remains to be seen. In our study, we found good brain–CNN correspondence between lower CNN layers and lower visual areas for processing real-world objects. Thus, the mixing of the different features in lower CNN layers is well-matched with that of lower visual areas. Yet these lower CNN layers fail to capture lower visual areas' responses for artificial objects. This indicates that some fundamental differences exist between the brain and CNNs at lower levels of visual processing that may not be overcome by remixing the CNN features.

What could be driving the difference between the brain and CNNs in visual processing? In recent studies, Baker et al.[44] and Geirhos et al.[31,52] reported that CNNs rely on local texture and shape features rather than global shape contours. This may explain why in our study lower CNN layers were able to fully capture the representational structures of real-world object images in lower visual areas, as processing in these brain areas likely relies more on local contours and texture patterns given their smaller receptive field sizes. As high-level object vision relies more on global shape contour processing (e.g., [53]), the lack of such processing in CNNs may account for CNNs' inability to fully capture processing in higher visual areas. This can be seen more directly in higher-level representations of our artificial objects (which share similar texture and contour elements at the local level but differ in how these elements are conjoined at the local and global levels). Specifically, while the brain takes both local and global shape similarities into consideration when grouping these objects, CNNs may rely mainly on local shape similarities (see the MDS plots in Fig. 5 and Supplementary Fig. 17). At lower levels of visual processing, the human brain likely encodes both shape elements and how they are conjoined at the local level to help differentiate the different artificial objects. CNNs, on the other hand, may rely more on the presence/absence of a particular texture patch or a shape element than on how they are conjoined at the local level to differentiate these objects. This may account for the divergence between the brain and CNNs at lower levels of visual processing for these artificial objects. Training with stylized images did not appear to improve performance in Resnet-50, suggesting that the differences between CNNs and the human brain may not be overcome by this type of training.

In two other studies involving real-world object images, we found additional differences between the human brain and CNNs in the development of transformation tolerant visual representations and the relative coding strength of object identity and nonidentity features[54,55]. Forming transformation-tolerant object identity representation has been argued to be the hallmark of primate vision, as it reduces the complexity of learning by requiring much fewer training examples and with the resulting representations being more generalizable to new instances of an object (e.g., in different viewing conditions) and to new exemplars of a category not included in the training. It could potentially dictate how objects are organized in the representational space in the brain, as examined in this study. While the magnitude of invariant object representation increases from lower to higher visual areas in the human brain, in the same 14 CNNs tested here, such invariance actually goes down from lower to higher CNN

**Table 1 A summary of the experimental parameters.**

| | Number of participants | Stimuli | Conditions | ROIs | Number of voxels per ROI |
|---|---|---|---|---|---|
| Experiment 1 | 6 | Real-world objects (eight categories—bodies, cars, cats, chairs, elephants, faces, houses, and scissors) | Original and controlled | V1-V4, LOT and VOT | 75 |
| Experiment 2 | 11 | Real-world objects (six categories—bodies, cars, chairs, elephants, faces, houses) | Full SF, high SF, and low SF | V1-V4, LOT and VOT | 75 |
| Experiment 3 | 6 | Real-world objects (eight categories—bodies, cars, cats, chairs, elephants, faces, houses, and scissors) and artificial objects (nine categories of cubies, spikes, and smoothies) | Natural and artificial objects | V1-V4, LOT, and VOT | 75 |

layers[54]. With its vast computing power, CNNs likely associate different instances of an object via a brute force approach (e.g., by simply grouping all instances of an object encountered under the same object label) without necessarily preserving the relationships among the objects across transformations and forming transformation-tolerant object representations. This again suggests that CNNs use a fundamentally different mechanism to group objects and solve the object recognition problem compared to the primate brain. In another study[55], we documented the relative coding strength of object identity and nonidentity features during visual processing in the human brain and CNNs. We found that identity representation increased and nonidentity feature representation decreased along the ventral visual pathway. In the same 14 CNNs examined here, while identity representation increased over the course of visual processing, nonidentity feature representation showed an initial large increase followed by a decrease at later stages of processing, different from the brain responses. As a result, higher CNN layers deviated more from the corresponding brain regions than lower layers did in how object identity and nonidentity features are coded with respect to each other. This is consistent with the RDM comparison results reported in this study.

CNNs' success in object categorization and their response correspondence with the primate visual areas have opened the exciting possibility that perhaps we can use CNN modeling as a viable scientific method to study primate vision. Presently, the detailed computations performed by CNNs are difficult for humans to understand, rendering them poorly understood information processing systems[3,56]. By analyzing results from three fMRI experiments and comparing visual representations in the human brain with 14 different CNNs, we found that CNNs' performance is related to how they are built and trained: they are built following the known architecture of the primate lower visual areas and are trained with real-world object images. Consequently, the best performing CNNs (i.e., Alexnet, Googlenet, Squeezenet, and Vgg-16) are successful at fully capturing the visual representational structures of lower human visual areas during the processing of both original and filtered real-world images, but not those of higher human visual areas during the processing of these images or that of artificial images at either level of processing. The close brain–CNN correspondence found in earlier fMRI studies thus might have been overly optimistic by including only real-world objects (which CNNs are generally trained on) and testing on data with relatively lower power. When we expanded the comparisons here to a broader set of filtered real-world stimuli and to artificial stimuli as well as testing on brain data with a higher power, we see large discrepancies between the brain and CNNs at both lower and higher levels of visual processing. While CNNs are successful in object recognition, some fundamental differences likely exist between the human brain and CNNs and preclude CNNs from fully modeling

the human visual system at their current states. This is unlikely to be remedied by simply changing the training images, changing the depth of the network, and/or adding recurrent processing. But rather, some fundamental changes may be needed to make CNNs more brain-like. This may only be achieved by our continuous research effort on understanding the precise algorithms used by the primate brain in visual processing to further guide CNN model development.

## Methods

**fMRI experimental details**. Details of the fMRI experiments have been described in two previously published studies[21,22]. They are summarized here for the readers' convenience (see also Table 1).

Six, ten, and six healthy human participants with normal or corrected to normal visual acuity, all right-handed and aged between 18 and 35, took part in Experiments 1–3, respectively. The sample size for each fMRI experiment was chosen based on prior published studies (e.g., [57,58]). All participants gave their written informed consent before the experiments and received payment for their participation. The experiments were approved by the Committee on the Use of Human Subjects at Harvard University. Each main experiment was performed in a separate session lasting between 1.5 and 2 h. Each participant also completed two additional sessions for topographic mapping and functional localizers. MRI data were collected using a Siemens MAGNETOM Trio, A Tim System 3T scanner, with a 32-channel receiver array head coil. For all the fMRI scans, a T2*-weighted gradient echo pulse sequence with TR of 2 s and a voxel size of 3 mm × 3 mm × 3 mm was used. fMRI data were analyzed using FreeSurfer (surfer.nmr.mgh.harvard. edu), FsFast[59], and in-house MATLAB codes. FMRI data preprocessing included 3D motion correction, slice timing correction, and linear and quadratic trend removal. Following standard practice, a general linear model was applied to the fMRI data to extract beta weights as response estimates.

In Experiment 1, we used cut-out gray-scaled images from eight real-world object categories (faces, bodies, houses, cats, elephants, cars, chairs, and scissors) and modified them to occupy roughly the same area on the screen (Fig. 1b). For each object category, we selected ten exemplar images that varied in identity, viewpoint/orientation, and pose (for the animal categories) to minimize the low-level similarities among them (see Supplementary Fig. 1 for the full set of images used). In this and the two experiments reported below, objects were always presented at fixation, and object positions never varied. In the original image condition, unaltered images were shown. In the controlled image condition, images were shown with contrast, luminance, and spatial frequency equalized across all the categories using the SHINE toolbox[23] (see Fig. 1b). Participants fixated at a central red dot throughout the experiment. Eye-movements were monitored in all the fMRI experiments to ensure proper fixation.

During the experiment, blocks of images were shown. Each block contained a random sequential presentation of ten exemplars from the same object category. Each image was presented for 200 ms followed by a 600 ms blank interval between the images (Fig. 1a). Participants detected a one-back repetition of the exact same image. This task-focused participants' attention on the object shapes and ensured robust fMRI responses. However, similar visual representations may be obtained when participants attended to the color of the objects[60,61] (see also[62]). Two image repetitions occurred randomly in each image block. Each experimental run contained 16 blocks, one for each of the 8 categories in each image condition (original or controlled). The order of the eight object categories and the two image conditions were counterbalanced across runs and participants. Each block lasted 8 s and was followed by an 8-s fixation period. There was an additional 8-s fixation period at the beginning of the run. Each participant completed one scan session with 16 runs for this experiment, each lasting 4 min 24 s.

In Experiment 2, only six of the original eight object categories were used including faces, bodies, houses, elephants, cars, and chairs. Images were shown in 3 conditions: Full-SF, High-SF, and Low-SF. In the Full-SF condition, the full

spectrum images were shown without modification of the SF content. In the High-SF condition, images were high-pass filtered using an FIR filter with a cutoff frequency of 4.40 cycles per degree (Fig. 1b). In the Low-SF condition, the images were low-pass filtered using an FIR filter with a cutoff frequency of 0.62 cycles per degree (Fig. 1b). The DC component was restored after filtering so that the image backgrounds were equal in luminance. Each run contained 18 blocks, one for each of the category and SF condition combinations. Each participant completed a single scan session containing 18 experimental runs, each lasting 5 min. Other details of the experiment design were identical to that of Experiment 1.

In Experiment 3, we used unaltered images from both real-world and artificial object categories. The real-world categories were the same eight categories used in Experiment 1, with the exemplars varying in identity, viewpoint/orientation, and pose (for the animal categories) to minimize the low-level similarities among them. The artificial object categories were nine categories of computer-generated 3D shapes (ten images per category) adopted from Op de Beeck et al.[24] and shown in random orientations to increase image variation within a category and to match the image variation of the exemplars used for the real-world object categories (see Fig. 1b; for the full set of artificial object images used, see Supplementary Fig. 2). Each run of the experiment contained 17 stimulus blocks, one for each object category (either real-world or artificial). Each participant completed 18 runs, each lasting 4 min 40 s. Other details of the experiment design were identical to that of Experiment 1.

We examined responses from independent localized lower visual areas V1–V4 and higher visual processing regions LOT and VOT. V1–V4 were mapped with flashing checkerboards using standard techniques[63]. Following the detailed procedures described in Swisher et al.[64] and by examining phase reversals in the polar angle maps, we identified areas V1–V4 in the occipital cortex of each participant (see also ref. [65]) (Fig. 1c). To identify LOT and VOT, following Kourtzi and Kanwisher[66], participants viewed blocks of face, scene, object, and scrambled object images. These two regions were then defined as a cluster of contiguous voxels in the lateral and ventral occipital cortex, respectively, that responded more to the original than the scrambled object images (Fig. 1c). LOT and VOT loosely correspond to the location of LO and pFs[66–68] but extend further into the temporal cortex in an effort to include as many object-selective voxels as possible in occipito-temporal regions.

LOT and VOT included a large swath of the ventral and lateral OTC and likely overlapped to a great extent with regions selective for specific object categories, including faces, bodies or scenes. To understand how the inclusion of these category-specific regions may affect the brain–CNN correlation, we also constructed LOT and VOT ROIs without the category-selective voxels. This was done by testing the category selectivity of each voxel in these two ROIs using the data from the main experiment. Specifically, since there were at least 16 runs in each experiment, using paired t tests, we defined a LOT or a VOT voxel as face-selective if its response was higher for faces than for each of the other non-face categories at $p < 0.05$. Similarly, a voxel was defined as body-selective if its response was higher for the average of bodies, cats, and elephants (in Experiment 2, only the average of bodies and elephants was used as cats were excluded in the experiment) than for each of the non-body categories at $p < 0.05$. Finally, a voxel was defined as scene-selective if its response was higher for houses than for each of the other non-scene categories at $p < 0.05$. In this analysis, a given object category's responses in the different formats (e.g., original and controlled) were averaged together. Given that each experiment contained at least 16 runs, using the main experimental data to define the category-selective voxels in LOT and VOT is comparable to how these voxels are traditionally defined. We used a relatively lenient threshold hold of $p < 0.05$ here to ensure that we excluded any voxels that exhibited any category selectivity, even if this occurred just by chance.

To generate the fMRI response pattern for each ROI in a given run, we first convolved an 8-s stimulus presentation boxcar (corresponding to the length of each image block) with a hemodynamic response function to each condition; we then conducted a general linear model analysis to extract the beta weight for each condition in each voxel of that ROI. These voxel beta weights were used as the fMRI response pattern for that condition in that run. Following Tarhan and Konkle[69], we selected the top 75 most reliable voxels in each ROI for further analyses. This was done by splitting the data into odd and even halves, averaging the data across the runs within each half, correlating the beta weights from all the conditions between the two-halves for each voxel, and then selecting the top 75 voxels showing the highest correlation. This is akin to including the best units in monkey neurophysiological studies. For example, Cadieu et al.[10] only selected a small subset of all recorded single units for their brain–CNN analysis. We obtained the fMRI response pattern for each condition from the 75 most reliable voxels in each ROI of each run. We then averaged the fMRI response patterns across all runs and applied z-normalization to the averaged pattern for each condition in each ROI to remove amplitude differences between conditions and ROIs.

**CNN details**. We tested 14 CNNs in our analyses (see Supplementary Table 1). They included both shallower networks, such as Alexnet, VGG16, and VGG 19, and deeper networks, such as Googlenet, Inception-v3, Resnet-50, and Resnet-101. We also included a recurrent network, Cornet-S, that has been shown to capture the recurrent processing in macaque IT cortex with a shallower structure[12,19]. This CNN has been recently argued to be the current best model of the primate ventral

visual processing regions[19]. All the CNNs used were trained with ImageNet images[30].

To understand how the specific training images would impact CNN representations, besides CNNs trained with ImageNet images, we also examined Resnet-50 trained with stylized ImageNet images[31]. We examined the representations formed in Resnet-50 pretrained with three different procedures[31]: trained only with the stylized ImageNet Images (RN50-SIN), trained with both the original and the stylized ImageNet Images (RN50-SININ), and trained with both sets of images and then fine-tuned with the stylized ImageNet images (RN50-SININ-IN).

Following O'Connel & Chun[32], we sampled between 6 and 11 mostly pooling and FC layers of each CNN (see Supplementary Table 1 for the specific CNN layers sampled). Pooling layers were selected because they typically mark the end of processing for a block of layers when information is pooled to be passed on to the next block of layers. When there were no obvious pooling layers present, the last layer of a block was chosen. For a given CNN layer, we extracted the CNN layer output for each object image in a given condition, averaged the output from all images in a given category for that condition, and then z-normalized the responses to generate the CNN layer response for that object category in that condition (similar to how fMRI category responses were extracted). Cornet-S and the different versions of Resnet-50 were implemented in Python. All other CNNs were implemented in Matlab. The output from all CNNs was analyzed and compared with brain responses using Matlab.

**Comparing the representational structures between the brain and CNNs**. To determine the extent to which object category representations were similar between brain regions and CNN layers, we correlated the object category representational structure between brain regions and CNN layers. To do so, we obtained the RDM from each brain region by computing all pairwise Euclidean distances for the object categories included in an experiment and then taking the off-diagonal values of this RDM as the category dissimilarity vector for that brain region. This was done separately for each participant. Likewise, from the CNN output, we computed pairwise Euclidean distances for the object categories included in an experiment to form the RDM and then taking the off-diagonal values of this RDM as the category dissimilarity vector for that CNN layer. We applied this procedure to each sampled layer of each CNN.

We then correlated the category dissimilarity vectors between each brain region of each participant and each sampled CNN layer. Following Cichy et al.[5], all correlations were calculated using Spearman rank correlation to compare the rank order, rather than the absolute magnitude, of the category representational similarity between the brain and CNNs (see also ref. [33] similar results were obtained, however, when Pearson correlation was used instead, see the results reported in Supplementary Figs. 3, 6, 7, and 16). All correlation coefficients were Fisher z-transformed before group-level statistical analyses were carried out.

To evaluate the correspondence in representation between lower and higher CNN layers to lower and higher visual processing regions, for each CNN examined, we identified, in each human participant, the CNN layer that showed the best RDM correlation with each of the six brain regions included. We then assessed whether the resulting layer numbers increased from low to high visual regions using Spearman rank correlation. Finally, we tested the resulting correlation coefficients at the participant group level. If a close correspondence in representation exists between the brain and CNNs, the averaged correlation coefficients should be significantly above zero. All stats reported were from one-tailed t tests. One-tailed t tests were used here as only values above zero were meaningful. In addition, all stats reported were corrected for multiple comparisons for the number of comparisons included in each experiment using the Benjamini–Hochberg procedure with the false-discovery rate (FDR) controlled at $q = 0.05$[34].

To assess how successfully the category RDM from a CNN layer could capture the RDM from a brain region, we first obtained the reliability of the category RDM in a brain region across the group of human participants by calculating the lower and upper bounds of the noise ceiling of the fMRI data following the procedure described by Nili et al.[33]. Specifically, the upper bound of the noise ceiling for a brain region was established by taking the average of the correlations between each participant's RDM and the group average RDM including all participants, whereas the lower bound of the noise ceiling for a brain region was established by taking the average of the correlations between each participant's RDM and the group average RDM excluding that participant.

To evaluate the degree to which CNN category RDMs may capture those of the different brain regions, for each CNN, using one-tailed t tests, we examined how close the highest correlation between a CNN layer and a brain region was to the lower bound of the noise ceiling of that brain region. These correlation results are reported in Supplementary Figs. 6, 7, and 16. To transform these correlation results into the proportion of explainable brain RDM variance captured by the CNN, we divided the brain–CNN RDM correlation by the corresponding lower bound of the noise ceiling and then squared the resulting value. We evaluated whether a CNN could fully capture the RDM variance of a brain region by testing the difference between 1 and the highest proportion of variance captured by the CNN using one-tailed t tests. One-tailed t tests were used as only testing values below the lower bound of the noise ceiling (for measuring correlation values) or below 1 (for measuring the amount of variance captured) were meaningful here. The t test

results were corrected for multiple comparisons for the six brain regions included using the Benjamini–Hochberg procedure at $q = 0.05$. If a CNN layer was able to fully capture the representational structure of a brain region, then its RDM correlation with the brain region should exceed the lower bound of the noise ceiling of that brain region, and the proportion of variance explained should not differ from 1. Because the lower bound of the noise ceiling varied somewhat among the different brain regions, for illustration purposes, in Supplementary Figs. 6, 7, 16, and 19, we plotted the lower bound of the noise ceiling from all brain regions at 0.7 while maintaining the differences between the CNN and brain correlations with respect to their lower bound noise ceilings (i.e., by subtracting the difference between the actual noise ceiling and 0.7 from each brain–CNN correlation value). This did not affect any statistical test results.

To directly visualize the object representational structures in different brain regions and CNN layers, using classical multidimensional scaling, we placed the category RDMs onto 2D spaces with the distances among the categories approximating their relative similarities to each other. The same scaling factor was used to plot the MDS plot for each sampled layer of each CNN. Thus the distance among the categories may be directly compared across the different sampled layers of a given CNN and across CNNs. The scaling factor was doubled for the brain MDS plots for Experiments 1 and 3 and was quadrupled for Experiment 2 to allow better visibility of the different categories in each plot. Thus the distance among the categories may still be directly compared across the different brain regions within a given experiment and between Experiments 1 and 3. Since rotations and flips preserve distances on these MDS plots, to make these plots more informative and to see how the representational structure evolved across brain regions and CNN layers, we manually rotated and/or flipped each MDS plot when necessary. In some cases, to maintain consistency across plots, we arbitrarily picked a few categories as our anchor points and then rotated and/or flipped the MDS plots accordingly.

**Reporting summary**. Further information on research design is available in the Nature Research Reporting Summary linked to this article.

## Data availability
Data supporting the findings of this study are available at https://osf.io/tsz47/. Source data are provided with this paper.

## Code availability
Standard code from the listed software was used. No special code was developed for this study.

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

## Acknowledgements

We thank Martin Schrimpf for help implementing CORnet-S, JohnMark Tayler for extracting the features from the three Resnet-50 models trained with the stylized images, and Thomas O'Connell, Brian Scholl, JohnMark Taylor, and Nick Turk-Brown for helpful discussions and feedback on the results. The project is supported by NIH grants 1R01EY022355 and 1R01EY030854 to Y.X. M.V.P. was supported in part by NIH Intramural Research Program ZIA MH002035.

## Author contributions

The fMRI data used here were from two prior publications, with M.V.-P. and Y.X. designing the fMRI experiments and M.V.-P. collecting and analyzing the fMRI data. Y.X. conceptualized the present study and performed the brain–CNN correlation analyses reported here. Y.X. wrote the paper with comments from M.V.-P.

## Competing interests

The authors declare no competing interests.
