## [Peer Review File · Nature Communications]

REVIEWER COMMENTS

Reviewer #1 (Remarks to the Author):

The manuscript by Xu and Vaziri-Pashkam reanalyzes some previously acquired fMRI data in order to assess the accuracy of convolutional neural networks (CNNs) as models of the visual brain. Specifically, the authors take a large number of different CNNs and systematically compare the representations in these networks using RSA (representational similarity analysis) against different regions in human visual cortex. Importantly, the authors have high-quality datasets that were collected in response to a variety of natural and artificial objects.

Overall, the manuscript addresses an important and timely topic, and the contribution of a large systematic comparison on higher quality data (compared to previous publications) with specific image perturbations is quite valuable. Moreover, the paper is clear, comprehensive, and nicely executed.

I have some comments below, most of which I think the authors can fairly easily address, and I think doing so will strengthen the manuscript.

Perhaps the one shortcoming of the paper is a lack of specific insight into the aspects of the experimental data that are being missed by the CNNs (see Point 2 below). It may be possible to provide this with some effort, given the current state of the paper and the analyses. But I would like to stress that even without this, the paper is already strong enough to stand on its own.

Major comments:

1. IS IT JUST GLASS HALF EMPTY / HALF FULL. The major theme of the paper is careful quantification of RDM prediction accuracy. It should be clarified early on in the text that what the authors mean by "fully capture" is to reach the lower bound of the noise ceiling. The authors may want to consider spending some time in the Discussion about this criterion for evaluating models. For example, apologists for CNN models may simply state that they never thought it would explain 100% of the noise ceiling, and that what's more important is the qualitative match, or something like that. The authors may wish to consider explicitly spelling out the stance that the paper takes, explaining the importance of quantitative accuracy. The authors may also wish to emphasize the numbers that they found so that the issue being tackled here is not just a "glass half empty" vs. "glass half full" subjective discussion, but rather that the authors now provide actual performance numbers showing that CNNs fare poorly for specific types of stimuli explored by the authors. (One note that if the authors wish to report percentage values, in order to properly quantify variance, one needs to square correlation values. For example, an observed correlation of 0.5 where the noise ceiling is 0.7 is actually $0.5^2 / 0.7^2 * 100 = 51\%$ of the noise ceiling in terms of percentage of explainable variance that is actually explained.)

2. SPECIFIC EFFECTS. Perhaps not surprising given the current state of research, but the paper uses the common approach of large-scale data comparison using lots of correlations and what not. This, while valuable, fails to provide specific insights into whether or not CNN models account for, or fail to account for, specific effects in experimental data. This leaves the situation somewhat unsatisfying, or at least, the authors have the opportunity (given their data and analyses) to provide such insight. For example, the authors conclude that "some fundamental differences exist [between CNNs and the brain]" --- but wouldn't it be more satisfying to state what those differences actually are? One idea is to summarize the RDMs from the CNN models and/or summarize the RDMs from the brain data and to show some representative examples, visualize these RDMs, and show specific successes and failures of the models. If this is provided, the authors can not only conclude that CNNs fall short quantitatively, but can also show in what specific interpretable ways they fail.

3. CROSS-STIMULUS DISTANCES. It appears that the authors perform comparisons using pairwise distances computed between stimulus conditions within different groups (e.g. within the natural objects). Is there a reason that the full pairwise distances were not computed? (E.g., the distances between the natural objects and the controlled objects.) This might provide further experimental data (on which models can be evaluated), and it may be quite useful for creating a single, consolidated, comprehensive RDM matrix that could be visualized and interpreted across CNN models and the brain (see Point 2).

4. SUMMARY PLOTS. It is commendable that detailed figures are provided showing all models, all ROIs, etc. However, summary plots are also very valuable. I would strongly suggest creating an (honest) summary of the results so that the reader can get, in a single glance, a summary of how the various models perform.

Minor comments:

5. It would be useful in the Methods to report the total dimensionality of the relevant quantities (e.g. number of conditions, number of subjects, number of voxels, ROIs, etc.) across the various experiments, so that the reader has a better sense of the extensiveness of the model evaluation. (In fact, the authors may consider publicly releasing these RDM matrices, especially since they are not that bulky?)

6. Can the z-normalization procedure be clarified? Is it the case that initially, the activity patterns are in percent signal change units where 0 is baseline? And then, the transformation is to take each activity pattern (voxels x 1) and z-score it? If so, it may be useful to point out that Euclidean distance after pattern z-scoring is quite similar to the commonly used approach of just using Pearson's correlation (r) to quantify similarity of activity patterns.

7. In the abstract, the rationale of "due to their high object categorization capabilities" seems a bit off. It presumably is both the ability of CNNs to categorize well as well as their general correspondence to brain data that are interesting to researchers.

8. The abstract should probably mention (since it is important) that the approach chosen in the paper is to use the RSA approach to compare models to brain data.

9. Consider inserting some citations for the idea of using both natural and artificial stimuli to test and understand the visual system.

10. Consider inserting some citations for the idea that "rendering [CNNs] poorly understood information processing systems".

11. Please clarify that in Figure 2 that you are showing averaged ranks, and please indicate what the error bars represent.

12. Please clarify that in Figure 3, you are presumably scaling each column of data points such that the noise ceiling corresponds to 0.7 (as opposed to adding/subtracting).

13. There are a few spelling mistakes/typos: for example, "modal", "temporal" "Resnet-50", "structural".

=====

Reviewer #2 (Remarks to the Author):

The authors use representational similarity analysis of three BOLD fMRI datasets to evaluate many convolutional neural networks as models of the human brain. They find that while many of the models show a correspondence to the brain (early layers of the models best explain early stages of visual processing, and later layers best explain higher-order stages of processing), none can explain brain responses in higher-order visual areas, i.e. lateral occipitotemporal cortex or ventral occipito-temporal cortex up to the noise ceiling.

The authors do an admirable job of evaluating a wide range of CNN models, including shallow and deep networks and a recent network trained on re-textured images (Gierhos et al 2019). The relative performance of these models seems like an interesting topic for further quantitation and discussion.

It is also worthwhile to note the limits of CNNs models and to explore situations in which they fail or under-perform. The issue that the authors raise of the low noise ceiling in past work is valid and important, and perhaps under-appreciated in the field. The authors also claim that CNNs do not explain as much variance in the responses of artificial stimuli as they do in photographs of real objects. This is potentially noteworthy as well, but see below for technical issues with this claim.

The most salient argument in this paper is that CNN models do not explain 100% of the explainable variance in higher-order visual areas - i.e., that they are not perfect. I do not think this point merits the emphasis the authors place on it. The principal paper to claim that CNN models capture 100% of the variance in brain responses - i.e. all the variance up to the noise ceiling - was Khaligh-Razavi & Kriegeskorte (2014). To me, this claim was somewhat questionable in the first place, given the issues that the authors astutely raise in this paper about the low noise ceiling in that work, and particularly given the numerous other studies that the authors here cite which do not find perfect model performance for CNNs (far from it). But also - and critically - the Kriegeskorte lab has also retracted the claim that CNNs explain variance up to the noise ceiling, due to a mathematical error in the paper in question (see <https://www.biorxiv.org/content/10.1101/2020.03.23.003046v1>).

Thus, I don't think that many researchers would claim that CNNs are perfect models of the human visual system. However, as the authors' own data shows, CNN-based models do explain a substantial amount of variance in visual responses. Based on other work, it seems clear that they explain more response variance than any other class of image-computable model to date. So to emphasize that they are not perfect seems to be missing an important point: as far as testable quantitative models go (i.e. as far as models go), they are as good as it currently gets. I would encourage the authors to note this point somewhere in the manuscript. As the paper stands, the authors seem to be arguing to throw the baby out with the bathwater - they say: "We conclude that current CNNs may not serve as sound working models of the human visual system." This seems too strong a statement. Per the adage, all models (including CNNs) are wrong (in some ways). I think a fair assessment is that CNNs seem to be exactly "working models" - models that have some flaws and shortcomings but do a very good job of explaining brain responses relative to other models.

The authors' critique of CNN models also has some important limitations.

First, the authors mention "one approach using linear transformation to link individual fMRI voxels to the units of CNN layers through training and cross-validation". This approach has been called the encoding model approach, voxelwise modeling approach, or voxel receptive-field modeling approach. The generalization of this approach to RSA via "mixed" RSA (Khaligh-Razavi et al, 2017) has been shown to substantially improve the match of some models to brain representations. The introduction of this 2017 paper lays out the advantages of this method over standard RSA very clearly:

"RSA makes it easy to test 'fixed' models, that is, models that have no free parameters to be fitted. Fixed models can be obtained by optimizing parameters for task performance. This approach is essential, because realistic models of brain information processing have large numbers of parameters (reflecting the substantial domain knowledge required for feats of intelligence), and brain data are costly and limited. However, we may still want to adjust our models on the basis of brain data. For example, it might be that our model contains all the nonlinear features needed to perfectly explain a brain representation, but in the wrong proportions: with the brain devoting more neurons to some representational features than to others. Alternatively, some features might have greater gain than others in the brain representation. Both of these effects can be modelled by assigning a weight to each feature. If a fixed model's RDM does not match the brain RDM, it is important to find out whether it is just the feature weighting that is causing the mismatch."

One strong possibility is that CNNs are over-parameterized with respect to the brain signals measured by fMRI. This seems particularly likely to be true when considering only responses to six to nine categories of objects. Brain responses in a given voxel or region may be well-captured by only SOME of the channels in the CNN, but other channels may be unrelated to the brain responses measured in a given experiment. Thus, any approach that fits the model to the data - that assesses the importance of each CNN feature to each voxel - will allow for this possibility, and likely provide a better model of the brain, and a fairer assessment of the utility of CNN models. I would encourage the authors to try a model fitting approach. Mixed RSA seems a very straightforward choice. In Khaligh-Razavi et al 2017, they found that CNN models particularly benefited from mixed RSA. Thus it seems likely that remixing of features could improve model performance and affect their conclusions here.

Another important issue is that the plots in figure 5 appear to be misleading. Figure 5a shows the noise ceilings for the two data sets used in Experiment 3. The noise ceilings for the natural images are substantially higher than the noise ceilings for the artificial images: for the natural image stimuli, four of the ROIs (V1, V2, LOT, and VOT) have a lower bound of the noise ceiling that appears to be greater than 0.8, whereas for the artificial images, the lower bound of the noise ceilings for all ROIs appears to be less than 0.6. Yet for the other sections of Figure 5, the authors plot the same noise ceiling (0.7) for all plots. This makes the performance of the CNNs for artificial stimuli appear to be farther from the noise ceiling than they actually are. I do see the difficulty in plotting different noise ceilings for different ROIs on the same plot, but this decision seems problematic. One alternative would be to plot the performance for each model and ROI as a proportion of the appropriate noise ceiling. The authors should also provide a quantitative comparison to substantiate their claim that "CNNs performed much worse in capturing visual processing of artificial than real-world object images in the human brain."

One more small issue is the use of large ROIs for LOT and VOT. These large ROIs are consistent with spirit of the "human IT" ROI used in Khaligh-Razavi et al 2014; however, I would argue that they are not ideal. Both nearly undoubtedly contain parcels of cortex known to be face- and body-selective (FFA and FBA in the ventral ROI and EBA, OFA, and possibly pSTS in the lateral ROI). The presence of these ROIs within their larger ROIs likely makes responses to faces and animals very distinct from each other and from other classes of stimuli in the brain RDMs for LOT and VOT. It seems highly likely that with or without re-weighting, CNN RDMs might not capture this strongly

categorical distinction, but that re-weighting the CNN RDMs might allow them to capture this distinction quite well. Finer-grained ROIs would make the present results more interesting, even without re-weighting: they could show whether un-weighted CNN models provide a worse match for brain data specifically in strongly categorical areas, and perhaps a better match in areas apparently domain-general areas such as the posterior fusiform gyrus. As mentioned, the authors' ROI definitions are in keeping with the some of the past results they cite (though see ROI definitions in the 2017 paper). However, since the authors make a point of criticizing the lack of functionally defined ROIs in previous work, it seems fair to point out that they have not defined several functional ROIs that are almost certainly affecting their results.

We thank the reviewers for their time and effort and their detailed comments of the manuscript. We have addressed in detail each comment raised by the reviewers below and have revised the manuscript accordingly. For easy referencing, we have included the reviewers' original comments in the italicized text below. In the revised manuscript, we have used track change to mark all the changes that we have made to the revised manuscript.

Reviewer #1 (Remarks to the Author):

The manuscript by Xu and Vaziri-Pashkam reanalyzes some previously acquired fMRI data in order to assess the accuracy of convolutional neural networks (CNNs) as models of the visual brain. Specifically, the authors take a large number of different CNNs and systematically compare the representations in these networks using RSA (representational similarity analysis) against different regions in human visual cortex. Importantly, the authors have high-quality datasets that were collected in response to a variety of natural and artificial objects.

Overall, the manuscript addresses an important and timely topic, and the contribution of a large systematic comparison on higher quality data (compared to previous publications) with specific image perturbations is quite valuable. Moreover, the paper is clear, comprehensive, and nicely executed.

I have some comments below, most of which I think the authors can fairly easily address, and I think doing so will strengthen the manuscript.

Perhaps the one shortcoming of the paper is a lack of specific insight into the aspects of the experimental data that are being missed by the CNNs (see Point 2 below). It may be possible to provide this with some effort, given the current state of the paper and the analyses. But I would like to stress that even without this, the paper is already strong enough to stand on its own.

We thank the reviewer for the overall positive evaluation of the manuscript. We completely agree that the next step is to figure out exactly what is being missed by the CNNs. We have provided additional analysis to address this question as well as included the description of two other recently completed projects examining these data. They have yielded some interesting insights regarding how the brain and CNNs may differ. We describe the findings of these two projects in our reply to Point 3 below.

Major comments:

1. IS IT JUST GLASS HALF EMPTY / HALF FULL. The major theme of the paper is careful quantification of RDM prediction accuracy. It should be clarified early on in the text that what the authors mean by "fully capture" is to reach the

*lower bound of the noise ceiling. The authors may want to consider spending some time in the Discussion about this criterion for evaluating models. For example, apologists for CNN models may simply state that they never thought it would explain 100% of the noise ceiling, and that what's more important is the qualitative match, or something like that. The authors may wish to consider explicitly spelling out the stance that the paper takes, explaining the importance of quantitative accuracy. The authors may also wish to emphasize the numbers that they found so that the issue being tackled here is not just a "glass half empty" vs. "glass half full" subjective discussion, but rather that the authors now provide actual performance numbers showing that CNNs fare poorly for specific types of stimuli explored by the authors. (One note that if the authors wish to report percentage values, in order to properly quantify variance, one needs to square correlation values. For example, an observed correlation of 0.5 where the noise ceiling is 0.7 is actually $0.5^2 / 0.7^2 * 100 = 51\%$ of the noise ceiling in terms of percentage of explainable variance that is actually explained.)*

This is an excellent comment. CNNs' success in object categorization has generated the excitement that perhaps the algorithms essential for high-level primate vision would automatically emerge in CNNs to provide us with a shortcut to understand and model primate vision. However, if our ultimate goal is to fully understand primate vision using CNN modeling as a viable scientific method, then the question is not about glass full/half empty, but rather, it is about how far we can go with this method, and whether there are fundamental differences in visual processing between the brain and CNNs that would limit CNN modeling as a shortcut to understand primate vision. CNNs incorporates the known architectures of the primate early visual areas and then repeat this design motif multiple times in their designs. The present results show that while such a general design scheme and the training procedure involved are sufficient to fully capture low-level visual processing even under various image transformations, they are not able to fully capture high-level processing of real-world objects and the general processing of novel objects. We have added this discussion on the p.3 of the revised manuscript.

The idea of quantifying the results as the percentage of explainable variance is a very good one. We have replotted all the main results in this way. The correlation results and the noise ceiling plots originally shown in the main results are now included in the supplementary figures to facilitate the comparison between our results and prior correlation results. The highest amount of brain RDM variance from higher visual regions LOT and VOT that could be captured by CNNs is about 60%, on par with previous neurophysiological results from macaque IT cortex (Cadieu et al., 2014; Yamins et al., 2014; Kar et al. 2019; Bao et al., 2020).

2. SPECIFIC EFFECTS. Perhaps not surprising given the current state of research, but the paper uses the common approach of large-scale data comparison using lots of correlations and what not. This, while valuable, fails to provide specific insights into whether or not CNN models account for, or fail to

account for, specific effects in experimental data. This leaves the situation somewhat unsatisfying, or at least, the authors have the opportunity (given their data and analyses) to provide such insight. For example, the authors conclude that "some fundamental differences exist [between CNNs and the brain]" --- but wouldn't it be more satisfying to state what those differences actually are? One idea is to summarize the RDMs from the CNN models and/or summarize the RDMs from the brain data and to show some representative examples, visualize these RDMs, and show specific successes and failures of the models. If this is provided, the authors can not only conclude that CNNs fall short quantitatively, but can also show in what specific interpretable ways they fail.

We agree with the reviewer that showing that CNNs differ from the brain is not entirely satisfying. It would be important to understand what is exactly different. We have followed the reviewer's suggestions and now include MDS plots from the first two and the last two brain regions examined (i.e., V1, V2, LOT and VOT) and from the first two and the last two layers sampled from four example CNNs (i.e., Alexnet, Cornet-S, Googlenet, and Vgg-19) from Experiments 1 and 3 in Figure 5. MDS plots from all brain regions and CNN layers samples from all experiments are included in Supplementary Figures 6-10 and 15. Consistent with our quantitative analysis, for real-world object images, there are some remarkable brain-CNN similarities in the representational structure for low level object representations (such as in Alexnet and Googlenet), but not for high level object representations. For high level object representations, both the brain and CNNs show a broad distinction between animate and inanimate objects (i.e., bodies, cats, elephants and faces vs. cars, chairs, houses and scissors), but not how these categories are represented with respect to each other. As Reviewer 2 mentioned below, one source of such difference could be the inclusion of category selective voxels in VOT and LOT in the human brain, as CNNs may not have developed such specialized units during training. In a new analysis performed, we excluded such voxels from VOT and LOT but saw very little change in brain-CNN RDM correlations (see our more detailed reply to Reviewer 2 below). For artificial object images, differences in MDS plots from lower brain regions and lower CNN layers are not easily interpretable. The differences between higher brain regions and higher CNN layers suggest that while the brain takes both local and global shape similarity into consideration when grouping objects, CNNs rely mainly on local shape similarities. This is consistent with other findings that specifically manipulated local and global shape similarity.

In two recently completed projects, we have taken further steps to understand potentially quantitative differences between the CNNs and brain. See our reply to the next comment.

We have included the above discussion in the revised manuscript on pp.12-13 and p.15.

3. CROSS-STIMULUS DISTANCES. It appears that the authors perform comparisons using pairwise distances computed between stimulus conditions within different groups (e.g. within the natural objects). Is there a reason that the

full pairwise distances were not computed? (E.g., the distances between the natural objects and the controlled objects.) This might provide further experimental data (on which models can be evaluated), and it may be quite useful for creating a single, consolidated, comprehensive RDM matrix that could be visualized and interpreted across CNN models and the brain (see Point 2).

In our preliminary analysis, we have done exactly what the reviewer suggested, see an example MDS plot below:

By looking at the MDS plots generated from these comprehensive RDM matrices, what strikes us right away is that there are interesting aspects of the representational structure that deserve detailed analysis at multiple different levels. Directly correlating the comprehensive RDM matrices between the brain and CNNs would just give us a single number and may not tell us exactly what could be different between them. As such, besides the present manuscript, which exclusively compares the RDM similarity between the brain and CNNs for representing natural and artificial objects, we have recently completed two other projects. In one project (Xu & Vaziri-Pashkam, 2020a), we correlated RDMs between two values of a transformation (e.g., between the original and controlled images) for a given brain region or CNN layer and examined how RDM similarity would change from lower to higher visual areas/CNN layers. This is closely related to the formation of transformation tolerant visual representations in the brain. In the other project (Xu & Vaziri-Pashkam, 2020b), we examined whether an object is more similar to the other objects sharing the same image format (e.g., cats to other objects all appearing in the original image format) than to the same object in the other

image format (e.g., cats in the original image format to cats in the controlled image format). This allowed us to document the relative coding strength of object identity and nonidentity features in both the brain and CNNs, and how this may change across as visual information ascends the processing hierarchy. Object identity and nonidentity information has so far been studied in relative isolation. By documenting the relative coding strength of these two types of features and how it may change during visual processing, we can develop a new tool to characterize feature coding in the human brain and the correspondence between the brain and CNNs. Besides manipulating image stats (original vs controlled) and SF (high SF vs low SF), we have also included additional fMRI experiments manipulating position (top vs bottom) and size (small vs large). I describe the details of these two projects below.

In the first recently completed project, we examined the development of transformation tolerant visual representations. Forming transformation-tolerant object identity representation has been argued to be the hallmark of primate vision, as it reduces the complexity of learning by requiring much fewer training examples and with the resulting representations being more generalizable to new instances of an object (e.g., in different viewing conditions) and to new instances of a category not included in training. Existing single cell neural recording findings predict that, as information ascends the visual processing hierarchy in the primate brain, the relative similarity among the objects would be increasingly preserved across identity-preserving image transformations. Interestingly, this key prediction at the population level has never been directly tested. By examining two types of Euclidian transformations (position and size) and two types of non-Euclidian transformations (image stats and the spatial frequency content of an image), we confirm this prediction across the entire human ventral visual processing pathway. We then analyzed results from the same 14 CNNs tested here and found that CNNs do not form the same kind of invariant object representations during the course of their visual processing as the human brain does. If anything, the magnitude of invariance representation (i.e., RDM correlation between image formats) actually goes down from lower to higher CNN layers, the opposite of what we see in the human brain. We observed similar performance between shallow and deep CNNs (e.g., Alexnet vs Googlenet), and the recurrent CNN does not perform better than the other CNNs. With its vast computing power, CNNs likely associate different instances of an object via a brute force approach (e.g., by simply grouping all instances of an object encountered under the same object label) without preserving the relationships among the objects across transformations and forming transformation-tolerant object representations. This suggests that CNNs likely use a fundamentally different mechanism to organize objects in the representational space and solve the object recognition problem compared to the primate brain.

In the second recently completely project, we documented the relative coding strength of object identity and nonidentity features in a brain region and how this may change across the human ventral visual pathway. We examined the same four nonidentity features as in the last project, including two Euclidean features (position and size) and two non-Euclidean features (image statistics and spatial frequency content of an image). Overall, identity representation increased and nonidentity feature representation

decreased along the ventral visual pathway, with identity outweighing the non-Euclidean features, but not the Euclidean ones, in higher levels of visual processing. In the same 14 CNNs examined here, we found that while the relative coding strength of object identity and nonidentity features in lower CNN layers matched well with that in early human visual areas, the match between higher CNN layers and higher human visual regions were limited. This is consistent with the RDM correlation results we report in this manuscript.

We have included a brief description of these two recently completed projects in the revised manuscript on pp.21-22. Each of these projects taps into a unique and independent aspects of the representational space. We have originally thought about including all three projects in a single paper, but then realized that doing so would not have provided us with sufficient space to fully develop each project in depth.

4. SUMMARY PLOTS. It is commendable that detailed figures are provided showing all models, all ROIs, etc. However, summary plots are also very valuable. I would strongly suggest creating an (honest) summary of the results so that the reader can get, in a single glance, a summary of how the various models perform.

We have now included summary plots to summarize all the main results. Figure 7a summarizes the results from the six datasets for the real-world object images (i.e., results from Figures 2-5) and Figure 7b summarizes the results for the artificial objects (i.e., results from Figures 2c and 5b). In each plot, the black bar represents the proportion of the cases in which significant brain-CNN linear correlations were obtained (a marginally significant correlation was counted as half a case); and each of the colored bars represents the proportion of the cases that RDM variance from each brain region was fully captured by the CNN (here again, a marginally significant correlation was counted as half a case). Alexnet, Googlenet, Squeezenet and Vgg-16 showed the best brain-CNN correspondence overall among the 14 CNNs examined. These summary plots show that, for real-world objects, although many networks show a significant brain-CNN linear correlation, only a select few are able to consistently capture brain RDM variance in lower visual brain regions (i.e., V1 to V3). No CNN is able to capture brain RDM variance in higher visual regions (i.e., LOT and VOT) in any of the cases. For artificial objects, a large number of networks showed a significant brain-CNN linear correlation; however, among these CNNs, the networks that were able to fully capture brain RDM variance for real-world objects in lower visual regions all failed to capture that of the artificial objects in neither lower nor higher visual regions.

Minor comments:

5. It would be useful in the Methods to report the total dimensionality of the relevant quantities (e.g. number of conditions, number of subjects, number of voxels, ROIs, etc.) across the various experiments, so that the reader has a better sense of the extensiveness of the model evaluation. (In fact, the authors

may consider publicly releasing these RDM matrices, especially since they are not that bulky?)

We have added a table in Methods to summarize the parameters of all the experiments included.

We do plan to publicly releasing these RDM matrices once the final revisions of the paper are completed so we can include all the main data and supplementary data that are presented in the paper.

6. Can the z-normalization procedure be clarified? Is it the case that initially, the activity patterns are in percent signal change units where 0 is baseline? And then, the transformation is to take each activity pattern (voxels x 1) and z-score it? If so, it may be useful to point out that Euclidean distance after pattern z-scoring is quite similar to the commonly used approach of just using Pearson's correlation (r) to quantify similarity of activity patterns.

The z-normalization procedure is described in detailed in Methods. Specifically, for each ROI in a given run, we first convolved an 8-second stimulus presentation boxcar (corresponding to the length of each image block) with a hemodynamic response function to each condition; we then conducted a general linear model analysis to extract the beta weight for each condition in each voxel of that ROI. These voxel beta weights were used as the fMRI response pattern for that condition in that run. We then averaged the fMRI response patterns across all runs and applied z-normalization to the averaged pattern for each condition in each ROI to remove amplitude differences between conditions and ROIs.

It is indeed the case that Euclidean distance after pattern z-scoring is quite similar, although not identical as our results show, to using Pearson's correlation to quantify similarity of activity patterns. We have added this comment to the revised manuscript on p.11.

7. In the abstract, the rationale of "due to their high object categorization capabilities" seems a bit off. It presumably is both the ability of CNNs to categorize well as well as their general correspondence to brain data that are interesting to researchers.

That's a good point. We have corrected this statement to include both CNN's high object categorization performance and their general correspondence to brain data.

8. The abstract should probably mention (since it is important) that the approach chosen in the paper is to use the RSA approach to compare models to brain data.

We have added this information to the abstract. We agree that it would be informative and important to make this information explicit in the abstract.

9. Consider inserting some citations for the idea of using both natural and artificial stimuli to test and understand the visual system.

We have expanded the section on this topic and have added citations on pp.13-14 of the revised manuscript.

10. Consider inserting some citations for the idea that "rendering [CNNs] poorly understood information processing systems".

We have added Kay (2018) and Serre (2019) for this idea on p.22 of the revised manuscript.

11. Please clarify that in Figure 2 that you are showing averaged ranks, and please indicate what the error bars represent.

Figure 2 indeed shows the averaged ranks across the participants in a given experiment. The error bars represent standard errors of the mean rank across the participants. This clarification has been added to Figure 2 caption of the revised manuscript.

12. Please clarify that in Figure 3, you are presumably scaling each column of data points such that the noise ceiling corresponds to 0.7 (as opposed to adding/subtracting).

We actually used addition/subtraction to bring the noise ceiling to 0.7 for the different brain regions without scaling. Scaling would have resulted in error bars appearing in different sizes for the different brain regions. All of this was done for visualization only and did not change the reported stats. We have clarified this on the figure captions in the revised manuscript.

13. There are a few spelling mistakes/typos: for example, "modal", "temporal" "Resnet-50", "structural".

Our apologies for these spelling mistakes/typos. They have been corrected in the revised manuscript. We thank the reviewer for catching them.

Reviewer #2 (Remarks to the Author):

The authors use representational similarity analysis of three BOLD fMRI datasets to evaluate many convolutional neural networks as models of the human brain. They find that while many of the models show a correspondence to the brain (early layers of the models best explain early stages of visual processing, and

later layers best explain higher-order stages of processing), none can explain brain responses in higher-order visual areas, i.e. lateral occipitotemporal cortex or ventral occipito-temporal cortex up to the noise ceiling.

The authors do an admirable job of evaluating a wide range of CNN models, including shallow and deep networks and a recent network trained on re-textured images (Gierhos et al 2019). The relative performance of these models seems like an interesting topic for further quantitation and discussion.

It is also worthwhile to note the limits of CNNs models and to explore situations in which they fail or under-perform. The issue that the authors raise of the low noise ceiling in past work is valid and important, and perhaps under-appreciated in the field. The authors also claim that CNNs do not explain as much variance in the responses of artificial stimuli as they do in photographs of real objects. This is potentially noteworthy as well, but see below for technical issues with this claim.

The most salient argument in this paper is that CNN models do not explain 100% of the explainable variance in higher-order visual areas - i.e., that they are not perfect. I do not think this point merits the emphasis the authors place on it. The principal paper to claim that CNN models capture 100% of the variance in brain responses - i.e. all the variance up to the noise ceiling - was Khaligh-Razavi & Kriegeskorte (2014). To me, this claim was somewhat questionable in the first place, given the issues that the authors astutely raise in this paper about the low noise ceiling in that work, and particularly given the numerous other studies that the authors here cite which do not find perfect model performance for CNNs (far from it). But also - and critically - the Kriegeskorte lab has also retracted the claim that CNNs explain variance up to the noise ceiling, due to a mathematical error in the paper in question (see <https://www.biorxiv.org/content/10.1101/2020.03.23.003046v1>).

Thus, I don't think that many researchers would claim that CNNs are perfect models of the human visual system. However, as the authors' own data shows, CNN-based models do explain a substantial amount of variance in visual responses. Based on other work, it seems clear that they explain more response variance than any other class of image-computable model to date. So to emphasize that they are not perfect seems to be missing an important point: as far as testable quantitative models go (i.e. as far as models go), they are as good as it currently gets. I would encourage the authors to note this point somewhere in the manuscript. As the paper stands, the authors seem to be arguing to throw the baby out with the bathwater - they say: "We conclude that current CNNs may not serve as sound working models of the human visual system." This seems too strong a statement. Per the adage, all models (including CNNs) are wrong (in some ways). I think a fair assessment is that CNNs seem to be exactly "working models" - models that have some flaws and shortcomings but do a very good job of explaining brain responses relative to other models.

It is indeed worth noting and celebrating that CNNs can capture some aspects of brain responses, better than any other models that have been developed so far. The central question that the present study tries to address is not whether CNNs can capture brain responses, but rather what comes next: how far can we go to with this method as a viable scientific method to understand the primate brain? Would it provide us a shortcut to fully understand primate vision or could it potentially lead us down the wrong path? The present study shows that CNNs' performance is related to how they are built and trained: they are built following the known architecture of the primate early visual areas and are trained with real-world object images. Consequently, the best performing CNNs (i.e., Alexnet, Googlenet, Squeezenet and Vgg-16) are successful in exactly that: capable of explaining all the variances in lower visual areas during the processing of real-world objects, but not those associated with the processing of real-world objects in higher visual areas, or that associated with artificial objects at either levels of processing. The present results show that some fundamental differences exist between the human brain and CNNs and preclude CNNs from fully modeling the human visual system at their current states.

We have added the above discussion to the revised manuscript on p.3 to further clarify the main point of the present study.

The authors' critique of CNN models also has some important limitations.

First, the authors mention "one approach using linear transformation to link individual fMRI voxels to the units of CNN layers through training and cross-validation". This approach has been called the encoding model approach, voxelwise modeling approach, or voxel receptive-field modeling approach. The generalization of this approach to RSA via "mixed" RSA (Khaligh-Razavi et al, 2017) has been shown to substantially improve the match of some models to brain representations. The introduction of this 2017 paper lays out the advantages of this method over standard RSA very clearly:

"RSA makes it easy to test 'fixed' models, that is, models that have no free parameters to be fitted. Fixed models can be obtained by optimizing parameters for task performance. This approach is essential, because realistic models of brain information processing have large numbers of parameters (reflecting the substantial domain knowledge required for feats of intelligence), and brain data are costly and limited. However, we may still want to adjust our models on the basis of brain data. For example, it might be that our model contains all the nonlinear features needed to perfectly explain a brain representation, but in the wrong proportions: with the brain devoting more neurons to some representational features than to others. Alternatively, some features might have greater gain than others in the brain representation. Both of these effects can be modelled by assigning a weight to each feature. If a fixed model's RDM does not match the brain RDM, it is important to find out whether it is just the feature weighting that is causing the mismatch."

One strong possibility is that CNNs are over-parameterized with respect to the brain signals measured by fMRI. This seems particularly likely to be true when considering only responses to six to nine categories of objects. Brain responses in a given voxel or region may be well-captured by only SOME of the channels in the CNN, but other channels may be unrelated to the brain responses measured in a given experiment. Thus, any approach that fits the model to the data - that assesses the importance of each CNN feature to each voxel - will allow for this possibility, and likely provide a better model of the brain, and a fairer assessment of the utility of CNN models. I would encourage the authors to try a model fitting approach. Mixed RSA seems a very straightforward choice. In Khaligh-Razavi et al 2017, they found that CNN models particularly benefited from mixed RSA. Thus it seems likely that remixing of features could improve model performance and affect their conclusions here.

Khaligh-Razavi et al. (2017) used the data from Kay et al. (2008) in which the training set contained 1750 unique images, with each image shown twice, and the testing set contained 120 unique images, with each shown 13 times. Because we could only extract the averaged responses from each object category, depending on the exact experiment, we had only between 16 and 18 different stimulus conditions, with each shown between 16 to 18 times. We are thus vastly underpowered to perform the mixed RSA analysis and would not be able to provide an objective evaluation of the mixed RSA approach.

In Khaligh-Razavi et al. (2017), while CNN layers all showed improved brain RDM prediction with mixing, those of the unsupervised models did not. The authors stated that this is because “(t)he benefit of linear fitting of the representational space is therefore outweighed by the cost to prediction performance of overfitting” (p. 193). V1 and Hmax models in particular showed significantly decreased RDM prediction with mixing. However, these unsupervised models contained 4,000 or less features, whereas layers L1 to L4 of the CNN model contained greater than 43,000 features, L5 to L7 contained more than 4,000 features and L8 contained 1000 features. It is unclear why overfitting during training specifically harmed V1 and Hmax models but not any of the CNN layers. It would be useful to fully understand the balance between decreased model performance due to overfitting and increased model performance due to feature mixing, as well as the minimum amount of data needed for training and testing, to replicate Khaligh-Razavi et al. (2017) and verify the finding. More research is thus needed to verify the mixed RSA approach.

In our study, we found that lower layers in a number of CNNs could fully capture the brain variance in lower visual areas when processing real-world objects. This shows that the mixing of the different CNN features is well matched to predict responses in lower visual regions. However, these lower CNN layers still fail to capture brain variance during the processing of artificial objects. This cannot be explained by a less optimal mixing of CNNs feature to capture brain responses, but rather points to some fundamental differences between the brain and CNNs even during lower stages of visual processing.

We have added the above discussion in the revised manuscript on pp.19-20.

Another important issue is that the plots in figure 5 appear to be misleading. Figure 5a shows the noise ceilings for the two data sets used in Experiment 3. The noise ceilings for the natural images are substantially higher than the noise ceilings for the artificial images: for the natural image stimuli, four of the ROIs (V1, V2, LOT, and VOT) have a lower bound of the noise ceiling that appears to be greater than 0.8, whereas for the artificial images, the lower bound of the noise ceilings for all ROIs appears to be less than 0.6. Yet for the other sections of Figure 5, the authors plot the same noise ceiling (0.7) for all plots. This makes the performance of the CNNs for artificial stimuli appear to be farther from the noise ceiling than they actually are. I do see the difficulty in plotting different noise ceilings for different ROIs on the same plot, but this decision seems problematic. One alternative would be to plot the performance for each model and ROI as a proportion of the appropriate noise ceiling. The authors should also provide a quantitative comparison to substantiate their claim that "CNNs performed much worse in capturing visual processing of artificial than real-world object images in the human brain."

We actually shifted both the correlation plots and the lower bound of noise ceilings together in these plots so that the noise ceilings from the different brain regions would line up together at 0.7. In other words, we did not change the differences between the correlation plots and their corresponding noise ceilings. This was done purely for illustration purposes and did not affect the actual stats. We apologize that this was not made clear in the original manuscript. In the revised manuscript, we have further clarified this and have moved these plots to supplementary figures. In the revised main figures, we have instead plotted the proportion of variance explained as both reviewers suggested.

In the revised manuscript on pp.14-15, we have also provided stats to quantify the significant drop in the amount of variance captured by the CNNs to substantiate our statement that "CNNs performed much worse in capturing visual processing of artificial than real-world object images in the human brain".

One more small issue is the use of large ROIs for LOT and VOT. These large ROIs are consistent with spirit of the "human IT" ROI used in Khaligh-Razavi et al 2014; however, I would argue that they are not ideal. Both nearly undoubtedly contain parcels of cortex known to be face- and body-selective (FFA and FBA in the ventral ROI and EBA, OFA, and possibly pSTS in the lateral ROI). The presence of these ROIs within their larger ROIs likely makes responses to faces and animals very distinct from each other and from other classes of stimuli in the brain RDMs for LOT and VOT. It seems highly likely that with without re-weighting, CNN RDMs might not capture this strongly categorical distinction, but that re-weighting the CNN RDMs might allow them to capture this distinction quite well. Finer-grained ROIs would make the present results more interesting,

even without re-weighting: they could show whether un-weighted CNN models provide a worse match for brain data specifically in strongly categorical areas, and perhaps a better match in areas apparently domain-general areas such as the posterior fusiform gyrus. As mentioned, the authors' ROI definitions are in keeping with the some of the past results they cite (though see ROI definitions in the 2017 paper). However, since the authors make a point of criticizing the lack of functionally defined ROIs in previous work, it seems fair to point out that they have not defined several functional ROIs that are almost certainly affecting their results.

The reviewer raised a very valid point here. To understand how the inclusion of category specific regions may affect brain-CNN correlation, we also constructed LOT and VOT ROIs without category-selective voxels. This was done by testing category selectivity of each voxel in these two ROIs using the data from the main experiment. Specifically, since there were at least 16 runs in each experiment, using paired t-tests, we defined a LOT or a VOT voxel as face selective if its response was higher for faces than for each of the other non-face categories at $p < .05$. Similarly, a voxel was defined as body selective if its response was higher for the average of bodies, cats and elephants (in Experiment 2, only the average of bodies and elephants was used as cats were not included in the experiment) than for each of the non-body categories at $p < .05$. Finally, a voxel was defined as scene selective if its response was higher for houses than for each of the other non-scene categories at $p < .05$. In this analysis, a given object category's responses in the different formats (e.g., original and controlled) were averaged together. Given that each experiment contained at least 16 runs, using the main experimental data to define the category selective voxels in LOT and VOT is comparable to how these voxels are traditionally defined.

In most cases, the amount of brain RDM variance that could be capture by CNNs remain largely unchanged whether or not category selective voxels were included or excluded (see Supplementary Figures 11 and 12). Significant differences were observed in only 6% of the comparisons ($ps < .05$, uncorrected, see the caption of Supplementary Figures 11 and 12 for a list of these cases). However, even in these cases, the maximum amount of LOT and VOT RDM variance captured by CNNs was still significantly less than 1 ($ps < .05$, corrected). As such, the failure of CNNs to fully capture brain RDM at higher levels of visual processing is not due to the presence of category selective voxels in LOT and VOT.

REVIEWER COMMENTS

Reviewer #1 (Remarks to the Author):

The authors have done a thorough job in addressing my comments. The additional MDS analysis and discussion are valuable. These changes have strengthened the paper.

I have a few, somewhat minor comments that I hope the authors will address. Overall, I am satisfied with the paper.

MINOR COMMENTS:

1. I think the discussion added to the manuscript has been helpful in clarifying the overall stance of the paper. However, I suggest that a little bit additional explicit discussion would be useful. It seems that the authors' stance is that the broad, overarching inference that we should take from their results regarding the accuracy of CNN models in matching the brain is not merely quantitative mismatch, but rather there is a fundamental qualitative mismatch, and this is revealed by the authors' use of controlled, manipulated, and artificial stimuli. So, a suggestion is to take the text starting with "We found that CNNs' performance" on p. 5 (which feels more like interpretation and so less suitable for an Introduction) and move this to the discussion in the last paragraph of p. 22. There, perhaps the authors can emphasize that they are suggesting that their results point to fundamental problems with the current state of CNN models and this is revealed by the authors' use of manipulated stimuli. The idea could be something like: the correspondence found in earlier studies might have been overly optimistic given the use of only real-world objects (which CNNs are generally trained on), whereas if you stress the comparison (using artificial and/or filtered stimuli), then failures become apparent. And this is the primary novelty of the work. If this is in the ballpark of how the authors think about their results, then I think it would strengthen the impact of the paper on the reader by stating these ideas clearly and explicitly.

2. Regarding the MDS analyses, can you provide more methodological details on the particular flavor of MDS that you used and/or the parameters? This is necessary in order to interpret the plots. For example, was there any attempt to match the MDS plots across cases? (Rotations and flips preserve distances on the 2D visualizations; thus, one can consider deliberately applying such flips to make the figures more comparable.) As another example, are the distances shown in the MDS plots intended to be interpretable (e.g. so that we can compare across layers of a CNN)? Also, as a minor note, it is somewhat inaccurate to say "projected the first two dimensions" since that implies that there are some underlying two dimensions in the original distance matrix (as there would be in PCA)...rather, in most forms of MDS, the two dimensions that come out in the embedding space are constructed and are dependent on some choices of parameters (rather than extracted). Finally, I understand that the MDS plots are "icing on the cake" and not the main thrust of the manuscript, but could you at least describe one specific example of a difference that you find interesting and informative in the MDS plots (e.g. differences between high-level brain representations and CNN representations) (perhaps using a little arrowhead in the figure to highlight a data point or two)?

Reviewer #2 (Remarks to the Author):

The authors have clearly put a lot of work into the revision, and I think the paper has been improved. I still have a mixed reaction to the paper on the whole, though.

On one hand, the authors have added interesting analyses and done a lot of technically correct work to improve the paper. They have clearly described what they have done, and added some important caveats and discussion.

On the other hand, I would argue that some of the experimental design and analysis choices are still not optimal to test the correspondence between DNNs and the brain.

I would like to be clear that I think my objections are matters of opinion, and I do not want to reject a paper over concerns/opinions with which some people might reasonably disagree. But I also feel that it is important to state my concerns, as I do think that they bear on the strength of the conclusions.

I leave the decision of how to weight these strengths and concerns to the editor.

Strengths of the revised manuscript:

The new MDS analysis is interesting, sheds some light on what aspects of the representation are captured by each region and each layer within the various DNNs. For example, it seems clear that among the natural images, for VOT and LOT, the differences between faces, bodies, and houses are among the most pronounced (note the spread of dark red, purple, and blue dots), whereas for later layers of DNNs, houses are distinct from faces and bodies but faces and bodies are represented similarly to each other. The authors may want to make note of these specific points. Worth noting also may be that DNNs generally don't see disembodied heads or headless bodies in their training data, and so are not likely to have distinct representations of heads and bodies. In general, this result seems likely to be due to the fact that LOT and VOT contain specific regions that are very selective for faces, bodies, and places (OFA, EBA, and OPA in LOT, and FBA, FFA, and PPA in VOT). It would be interesting to see the MDS plots with the category-selective voxels excluded.

The analysis that removes category-selective voxels is also interesting. I am somewhat surprised, given the clear indications of selectivity for faces, bodies, and places in the MDS analysis, that this did not have a bigger effect on more of the networks. It is interesting to note that at least some networks did have a substantial gain in their match to the brain when category-selective voxels were excluded (the RDM-RDM correlation for AlexNet for high-frequency images, in particular, seemed to have increased from about 0.4 to 0.7 - a substantial increase). To me, it's a very interesting question why some networks increase their match to the brain and others do not. Might the size of the output for the layers in each network matter? I suspect that the size of the network may matter, because larger networks may have more units responsive to features that were not present in the conditions shown in this study. See the section below on mixed RSA analysis for more discussion related to this point. Another analysis that might be interesting - and I do not consider this requisite for this paper, to be clear - would be to simply drop either the face or body category from the natural images and test the correspondence of the slightly smaller DNN and brain RDMs.

The caveats the authors have added also help to clarify the intent and findings of the study.

Weaknesses of the revised manuscript:

While the caveats do help, I do not think that the summary figure provides a fair summary of the results. Figure 7 applies a strong threshold to the results: anything less than 100% of the variance is not included, and by implication is considered a failure. There is nothing technically wrong with this, as the authors have clearly explained what they have done. However, this once again seems to me to be throwing the baby out with the bathwater. Perfection seems a very harsh standard; to me, it seems harsh to the point of misleading, in that it equates explaining 70% of the potentially explainable variance with 0% of the potentially explainable variance. I think including mean performance of models in this figure would be more straightforward.

Regarding mixed RSA: I apologize for not realizing that the study would be under-powered to use mixed RSA. This should have been clear, given the number of data points available in the experiments. It is nonetheless disappointing that the authors are not able to perform an important test of brain-CNN correspondence given the limits of their design. To my mind, this leaves open the (likely) possibility that there may be a sufficient set features within each DNN model to accurately predict brain responses in these stimuli, but that the DNN models are simply over-parameterized given the relative simplicity of the stimuli - particularly for the artificial images, where the deep networks appear to match the brain most poorly. The authors appropriately note this limitation, which is to the good, but to me this still seems to weaken the result.

The authors have increased their focus on the mismatch between DNNs and the brain in lower-level areas (and layers). I think this emphasis and the analyses supporting it are problematic. In their rebuttal, the authors argue:

"In our study, we found that lower layers in a number of CNNs could fully capture the brain variance in lower visual areas when processing real-world objects. This shows that the mixing of the different CNN features is well matched to predict responses in lower visual regions. However, these lower CNN layers still fail to capture brain variance during the processing of artificial objects. This cannot be explained by a less optimal mixing of CNNs feature to capture brain responses, but rather points to some fundamental differences between the brain and CNNs even during lower stages of visual processing."

Maybe. However, it may also point to experimental design differences between the natural and artificial images. An important consideration when comparing brain responses to DNN representations in these two data sets is the low-level feature variability within each object category. Based on the description of the stimuli for each experiment in the methods, it sounds like there may have been more position and orientation variability in the artificial images. The authors write: "The real-world categories were the same eight categories used in Experiment 1. The artificial object categories were nine categories of computer-generated 3D shapes (ten images per category) adopted from Op de Beek et al. (2008) and shown in random orientations to increase image variation within a category (see Figure 5A)." To my reading, this sounds as if the artificial objects were varied in pose, but the real objects were all presented upright and (presumably) in largely overlapping positions across exemplars. If this is the case, it may explain the substantially higher noise ceiling in early visual areas for natural vs artificial images. More variability in one condition (artificial images) compared to another makes the results less clear to interpret.

Averaging across different orientations of images likely works against the DNN models. Early stages of the visual system are known to be driven by image contrast at particular locations and orientations. Thus, early visual areas should be expected to respond substantially differently to different orientations and positions of objects. For example, if an object is presented angled toward the lower left visual field vs toward the lower right, this difference is likely to be larger than any feature-specific difference (e.g. pointiness or roundness of parts of the objects), due to the retinotopic organization of V1-V4. Position variation would likely be captured by early layers of a

DNN model. But the design of the experiment averages across this variation. Thus, DNNs must predict the average response to many different positions to be considered a good model in this experiment. This seems like a sub-optimal test, as if the scales have already been tipped against the DNNs (at least in early visual areas) by the experimental design. This problem is worse to the extent that there is as much local image contrast variability within as between conditions.

Some variant of this problem may also exist for higher-level areas as well, though this is less clear-cut. In any case, DNNs are not given a chance to predict real differences between responses to different exemplars of a class, if such differences do exist. The mean response to many different exemplars is the only quantity that enters the RSA analysis. Thus, these experiments constitute a test of whether the full representation in DNNs (and not some subset of the units in each DNN) capture the coarse categorical structure of the representation that is consistent across subjects. This is not what I would consider to be the best way to test DNNs. But again, the authors have been pretty clear about what they have done and the limits of what they have done. So my objection is a matter of opinion, on which reasonable people may differ.

In terms of concrete changes I think should still be made: I think the authors should clarify whether the natural images were varied in orientation as the artificial ones were. If the natural images were not varied, then that difference should be clearly noted as a caveat to the difference in DNN performance in early visual areas. And I think the summary figure (Figure 7) would be better incorporating mean performance rather than whether DNNs captured 100% of the variance.

We thank the reviewers for their time and effort and their detailed comments on the manuscript. Below, we have addressed in detail each comment raised by the reviewers in the 2nd round of the review and have revised the manuscript accordingly. For easy referencing, we have included the reviewers' original comments in the italicized text below. In the revised manuscript, we have used track change to mark all the changes that we have made to the revised manuscript.

Reviewer #1 (Remarks to the Author):

The authors have done a thorough job in addressing my comments. The additional MDS analysis and discussion are valuable. These changes have strengthened the paper.

I have a few, somewhat minor comments that I hope the authors will address. Overall, I am satisfied with the paper.

We thank the reviewer for the overall positive evaluation of the manuscript and all the insightful comments provided.

MINOR COMMENTS:

1. I think the discussion added to the manuscript has been helpful in clarifying the overall stance of the paper. However, I suggest that a little bit additional explicit discussion would be useful. It seems that the authors' stance is that the broad, overarching inference that we should take from their results regarding the accuracy of CNN models in matching the brain is not merely quantitative mismatch, but rather there is a fundamental qualitative mismatch, and this is revealed by the authors' use of controlled, manipulated, and artificial stimuli. So, a suggestion is to take the text starting with "We found that CNNs' performance" on p. 5 (which feels more like interpretation and so less suitable for an Introduction) and move this to the discussion in the last paragraph of p. 22. There, perhaps the authors can emphasize that they are suggesting that their results point to fundamental problems with the current state of CNN models and this is revealed by the authors' use of manipulated stimuli. The idea could be something like: the correspondence found in earlier studies might have been overly optimistic given the use of only real-world objects (which CNNs are generally trained on), whereas if you stress the comparison (using artificial and/or filtered stimuli), then failures become apparent. And this is the primary novelty of the work. If this is in the ballpark of how the authors think about their results, then I think it would strengthen the impact of the paper on the reader by stating these ideas clearly and explicitly.

Following the reviewer's suggestion, we have now included additional explicit statements in the last paragraph of Discussion on pp.24-25 to clearly state that we

believe there are some fundamental qualitative differences between the human brain and CNNs. We thank the reviewer for this suggestion.

2. Regarding the MDS analyses, can you provide more methodological details on the particular flavor of MDS that you used and/or the parameters? This is necessary in order to interpret the plots. For example, was there any attempt to match the MDS plots across cases? (Rotations and flips preserve distances on the 2D visualizations; thus, one can consider deliberately applying such flips to make the figures more comparable.) As another example, are the distances shown in the MDS plots intended to be interpretable (e.g. so that we can compare across layers of a CNN)? Also, as a minor note, it is somewhat inaccurate to say "projected the first two dimensions" since that implies that there are some underlying two dimensions in the original distance matrix (as there would be in PCA)...rather, in most forms of MDS, the two dimensions that come out in the embedding space are constructed and are dependent on some choices of parameters (rather than extracted). Finally, I understand that the MDS plots are "icing on the cake" and not the main thrust of the manuscript, but could you at least describe one specific example of a difference that you find interesting and informative in the MDS plots (e.g. differences between high-level brain representations and CNN representations) (perhaps using a little arrowhead in the figure to highlight a data point or two)?

We thank the reviewer for all these great suggestions.

We used classical multidimensional scaling for the MDS plots. The same scaling factor was used to plot the MDS plot for each sampled layer of each CNN. Thus the distance among the categories may be directly compared across the different sampled layers of a given CNN and across CNNs. The scaling factor was doubled for the brain MDS plots for Experiments 1 and 3 and was quadrupled for Experiment 2 to allow better visibility of the different categories in each plot. Thus the distance among the categories may still be directly compared across the different brain regions within a given experiment and between Experiments 1 and 3. We have added this detail on p.34 of the revised manuscript.

In our last submission, we did not rotate or flip the MDS plots across cases, but rather presented each plot as the script generated them. We agree with the reviewer that aligning the MDS plots has the added benefit of making the plots more comparable between adjacent brain regions or sampled CNN layers and between the brain and CNNs, thereby making these plots more informative. In the revised MDS plots (both in the main figures and in the supplementary figures), we have inspected each MDS plot and rotated and/or flipped it when necessary. In some cases, we had to arbitrarily pick a few categories as our anchor points in order to create some consistency across the plots. We have added this detail on p.34 of the revised manuscript.

We thank the reviewer for pointing out the inaccuracy in our MDS description. We have reworded the text to make it more accurate as follows: "we placed the RDMs onto 2D

spaces with the distances among the categories approximating their relative similarities to each other". We have made this revision throughout the revised manuscript.

As the reviewer suggested, we have now put little dotted circles and dashed ovals around some data points in the MDS plots to highlight a few example differences between the brain and the CNNs in Figure 5. We then explain these examples in more detail in the main text on p.13 and p.16 and the caption of Figure 5 (we have considered using arrowheads as the reviewer suggested but found that they made the plots more cluttered and could be easily confused as data points).

Reviewer #2 (Remarks to the Author):

The authors have clearly put a lot of work into the revision, and I think the paper has been improved. I still have a mixed reaction to the paper on the whole, though.

On one hand, the authors have added interesting analyses and done a lot of technically correct work to improve the paper. They have clearly described what they have done, and added some important caveats and discussion.

On the other hand, I would argue that some of the experimental design and analysis choices are still not optimal to test the correspondence between DNNs and the brain.

I would like to be clear that I think my objections are matters of opinion, and I do not want to reject a paper over concerns/opinions with which some people might reasonably disagree. But I also feel that it is important to state my concerns, as I do think that they bear on the strength of the conclusions.

I leave the decision of how to weight these strengths and concerns to the editor.

Strengths of the revised manuscript:

The new MDS analysis is interesting, sheds some light on what aspects of the representation are captured by each region and each layer within the various DNNs. For example, it seems clear that among the natural images, for VOT and LOT, the differences between faces, bodies, and houses are among the most pronounced (note the spread of dark red, purple, and blue dots), whereas for later layers of DNNs, houses are distinct from faces and bodies but faces and bodies

are represented similarly to each other. The authors may want to make note of these specific points. Worth noting also may be that DNNs generally don't see disembodied heads or headless bodies in their training data, and so are not likely to have distinct representations of heads and bodies. In general, this result seems likely to be due to the fact that LOT and VOT contain specific regions that are very selective for faces, bodies, and places (OFA, EBA, and OPA in LOT, and FBA, FFA, and PPA in VOT). It would be interesting to see the MDS plots with the category-selective voxels excluded.

We agree with the reviewer's observation regarding the MDS plots. We also noted that within the inanimate objects, while the representations of cars, chairs, houses and scissors tend to form a square, those in higher CNN layers tend to form a line. We have added these observations in the revised manuscript on p.13. See also our reply to the 2nd comment of Reviewer 1.

CNNs are exposed to natural images just like the human visual system. Given that the human visual system generally do not see disembodies heads or headless bodies in its training data either, the goal of the study is not to test images that a system has been exposed to during training, but rather how it would react to images that it has not. If two systems are similar in their underlying representation, then they should still respond similarly to images that they are not exposed to during training. If not, then it unveils differences in how these two systems represent natural objects. This discussion has been added to the revised manuscript on p.14.

The reviewer raised an excellent point regarding what these MDS plots would look like when the category-selective voxels were excluded from higher visual regions. We have added these plots in the revised manuscript on Supplementary Figures 8-12 and 17. The plots look pretty much the same to our eyes with some very minor differences. This indicates that the RDM structure we have observed so far was not driven by the presence of the category specific areas in ventral visual cortex. But rather it persists even when voxels selective for these categories have been removed.

The analysis that removes category-selective voxels is also interesting. I am somewhat surprised, given the clear indications of selectivity for faces, bodies, and places in the MDS analysis, that this did not have a bigger effect on more of the networks. It is interesting to note that at least some networks did have a substantial gain in their match to the brain when category-selective voxels were excluded (the RDM-RDM correlation for AlexNet for high-frequency images, in particular, seemed to have increased from about 0.4 to 0.7 - a substantial increase). To me, it's a very interesting question why some networks increase their match to the brain and others do not. Might the size of the output for the layers in each network matter? I suspect that the size of the network may matter, because larger networks may have more units responsive to features that were not present in the conditions shown in this study. See the section below on mixed RSA analysis for more discussion related to this point. Another analysis that might be interesting - and I do not consider this requisite for this paper, to be clear - would be to simply drop

either the face or body category from the natural images and test the correspondence of the slightly smaller DNN and brain RDMS.

We believe what the reviewer refers to is the Full-SF, rather than the high SF condition in Experiment 2, which showed an increase in correlation for VOT in AlexNet from around 0.4 to 0.7 with the exclusion of the category selective voxels. The same Full-SF images were also shown in Experiment 1 Original condition and in Experiment 3 Natural condition. In Experiment 3 Natural condition, correlation for VOT in Alexnet actually went down from 0.65 to 0.4. So the improvement seen in Experiment 2 Full-SF condition in AlexNet was not replicated. We believe that this is likely a statistical fluke rather than a reliable improvement. We have added a short discussion in the revised manuscript on pp.13-14 regarding this finding.

The idea of removing faces or bodies from the natural images is an interesting one. Because the animal images we used also contain faces and bodies, to do this correctly, we would need to remove cats and elephants with faces if face images are to be removed, and remove these same two animal categories with bodies if body images are to be removed. One could even consider removing all the face and body images together to make a stronger case. All of these manipulations would unfortunately leave us with too few categories and insufficient power to conduct our analysis. It would be interesting in future research to more finely sample the categories, such as examining sufficient number of animate or inanimate categories alone, to see whether CNNs may do better for a subset of the real-world objects. That being said, in our study we have decided to take an equally valid approach by avoiding real-world objects all together and examining responses to artificial object categories instead. Here, no preexisting category information, semantic knowledge, as well as experience with the objects should affect object processing at the higher level, thereby putting the brain and CNN on even grounds. And yet, for the CNNs that are successful in fully tracking the lower but not higher level processing of real-world objects, they still perform poorly (if not worse) in tracking the processing of artificial objects at the higher level. This really shows that the discrepancy between the brain and CNNs at the higher level of visual processing is something more fundamental. We have added a summary of this discussion in the revised manuscript on p.15.

The caveats the authors have added also help to clarify the intent and findings of the study.

Weaknesses of the revised manuscript:

While the caveats do help, I do not think that the summary figure provides a fair summary of the results. Figure 7 applies a strong threshold to the results: anything less than 100% of the variance is not included, and by implication is considered a failure. There is nothing technically wrong with this, as the authors have clearly explained what they have done. However, this once again seems to me to be throwing the baby out with the bathwater. Perfection seems a very harsh standard; to me, it seems harsh to the point of misleading, in that it equates explaining 70%

of the potentially explainable variance with 0% of the potentially explainable variance. I think including mean performance of models in this figure would be more straightforward.

Our intention here was to show the success rate of CNNs in their abilities to fully explain the brain variance. But we agree with the reviewer that plotting the average performance of the CNNs would be a better choice. We have revised the figure and now show the average performance of the CNNs as well as the performance from each experiment on the same figure (see the revised Figure 8). We thank the reviewer for this suggestion.

Regarding mixed RSA: I apologize for not realizing that the study would be under-powered to use mixed RSA. This should have been clear, given the number of data points available in the experiments. It is nonetheless disappointing that the authors are not able to perform an important test of brain-CNN correspondence given the limits of their design. To my mind, this leaves open the (likely) possibility that there may be a sufficient set features within each DNN model to accurately predict brain responses in these stimuli, but that the DNN models are simply over-parameterized given the relative simplicity of the stimuli - particularly for the artificial images, where the deep networks appear to match the brain most poorly. The authors appropriately note this limitation, which is to the good, but to me this still seems to weaken the result.

As mentioned in our last response letter, it is unclear to us that the mixed RSA analysis is straightforward, as we don't fully understand the balance between decreased model performance due to overfitting and increased model performance due to feature mixing, as well as the minimum amount of data needed for training and testing. In our original fMRI experiments, we used a block design to increase power and obtained fairly high reliability across subjects, with the lower bound of the noise ceiling being around .8 for the unaltered real-world images. In comparison, noise ceiling for LO is just below .2 in Khaligh-Razavi et al. (2017). Thus the amount of explainable variance was less than 4%, which is really low. A mixed RSA approach requires brain responses from a large number of single images. This will necessarily result in lower power and lower reliability across subjects. In other words, due to noise, only a small amount of consistent neural responses are preserved across subjects, resulting in much of the neural data used to train the model likely just being subject-specific noise. This can significantly weaken the mixed RSA approach. Additionally, whether a mixed RSA model trained with one kind of object images (e.g., real-world object images) may accurately predict the responses from another kind of object images (e.g., artificial object images) has not been tested. Thus, although we agree with the reviewer that the general principle of a mixed RSA approach is promising, what it can actually deliver remains to be seen. We have added this discussion to the revised manuscript on p.22.

The authors have increased their focus on the mismatch between DNNs and the brain in lower-level areas (and layers). I think this emphasis and the analyses supporting it are problematic. In their rebuttal, the authors argue:

"In our study, we found that lower layers in a number of CNNs could fully capture the brain variance in lower visual areas when processing real-world objects. This shows that the mixing of the different CNN features is well matched to predict responses in lower visual regions. However, these lower CNN layers still fail to capture brain variance during the processing of artificial objects. This cannot be explained by a less optimal mixing of CNNs feature to capture brain responses, but rather points to some fundamental differences between the brain and CNNs even during lower stages of visual processing."

Maybe. However, it may also point to experimental design differences between the natural and artificial images. An important consideration when comparing brain responses to DNN representations in these two data sets is the low-level feature variability within each object category. Based on the description of the stimuli for each experiment in the methods, it sounds like there may have been more position and orientation variability in the artificial images. The authors write: "The real-world categories were the same eight categories used in Experiment 1. The artificial object categories were nine categories of computer-generated 3D shapes (ten images per category) adopted from Op de Beeck et al. (2008) and shown in random orientations to increase image variation within a category (see Figure 5A)." To my reading, this sounds as if the artificial objects were varied in pose, but the real objects were all presented upright and (presumably) in largely overlapping positions across exemplars. If this is the case, it may explain the substantially higher noise ceiling in early visual areas for natural vs artificial images. More variability in one condition (artificial images) compared to another makes the results less clear to interpret.

For the real-world object categories, we have actually used exemplars that varied in viewpoint/orientation and pose (for the animals). We wrote in Methods: "In Experiment 1, we used cut-out grey-scaled images from eight real-world object categories (faces, bodies, houses, cats, elephants, cars, chairs, and scissors) and modified them to occupy roughly the same area on the screen (Figure 1B). For each object category, we selected ten exemplar images that varied in identity, pose and viewing angle to minimize the low-level similarities among them." The same info also appeared in the caption of Figure 1. Due to this manipulation, we varied the orientations of the exemplars in each artificial category to make them more comparable to the exemplars shown in the real-world object categories. To avoid any future confusions, we have now included in Supplementary Figures 1 and 2 all the exemplars used in the real-world and artificial categories. We have also revised the details of Experiment 3 in Methods on p.28 to make our stimulus manipulation extra clear.

CNNs are trained to handle real-world objects varying in positions, viewpoints/orientations and poses (for the animals). This has been a big selling point for the CNNs. Thus the variability introduced in the artificial objects should not have posed a challenge to the CNNs. To us, their poor performance in handling the artificial objects

reflects some fundamental differences in how the brain and CNNs process objects, rather than the particularity of the stimulus set used.

The overall higher noise ceiling in early visual areas for natural than artificial images could be due to several reasons. First, visual neurons may be more responsive to natural than artificial objects, as our brain has evolved to process natural rather than artificial stimuli. Second, the natural images were overall more variable in identity within a category and thus more interesting to look at than the similar looking artificial images. This could potentially increase SNR for the natural objects. Lastly, the natural categories were more distinctive from each other than the artificial ones were. Thus any distortion in the representational structure due to noise would be smaller for the natural than for the artificial objects, potentially increasing RDM consistency among the subjects. We have added this discussion to the figure caption of Supplementary Figure 15 in the revised manuscript.

Averaging across different orientations of images likely works against the DNN models. Early stages of the visual system are known to be driven by image contrast at particular locations and orientations. Thus, early visual areas should be expected to respond substantially differently to different orientations and positions of objects. For example, if an object is presented angled toward the lower left visual field vs toward the lower right, this difference is likely to be larger than any feature-specific difference (e.g. pointiness or roundness of parts of the objects), due to the retinotopic organization of V1-V4. Position variation would likely be captured by early layers of a DNN model. But the design of the experiment averages across this variation. Thus, DNNs must predict the average response to many different positions to be considered a good model in this experiment. This seems like a sub-optimal test, as if the scales have already been tipped against the DNNs (at least in early visual areas) by the experimental design. This problem is worse to the extent that there is as much local image contrast variability within as between conditions.

Some variant of this problem may also exist for higher-level areas as well, though this is less clear-cut. In any case, DNNs are not given a chance to predict real differences between responses to different exemplars of a class, if such differences do exist. The mean response to many different exemplars is the only quantity that enters the RSA analysis. Thus, these experiments constitute a test of whether the full representation in DNNs (and not some subset of the units in each DNN) capture the coarse categorical structure of the representation that is consistent across subjects. This is not what I would consider to be the best way to test DNNs. But again, the authors have been pretty clear about what they have done and the limits of what they have done. So my objection is a matter of opinion, on which reasonable people may differ.

In the three experiments reported here, we always presented objects at fixation and never varied the positions of the objects. We have added this clarification to the revised manuscript on p.6 and p.26. Even though the exemplars from the real-world objects

varied in identity, viewpoint/orientation and pose (for the animals), several CNNs we tested were capable of fully capturing the representational structure of these object categories at the lower level of visual processing, even when filtered versions of the object images were shown. Thus, the concerns the reviewer raised may not be valid. These CNNs appear to be quite capable of handling exemplar variations and tracking brain responses at the lower level of visual processing. This suggests that CNNs' inability to track the lower level neural representations of artificial objects must be due to some other reasons, rather than the image variability we introduced among the exemplars.

Regarding the point on examining responses at the category rather than at the individual exemplar level, we have in fact already addressed this concern in Discussion in our original submission (see pp.19-20 of the revised manuscript): "Although we examined object category responses averaged over multiple exemplars rather than responses to each object, previous research has shown similar category and exemplar response profiles in macaque IT and human lateral occipital cortex with more robust responses for categories than individual exemplars due to an increase in SNR (Hung et al., 2005; Cichy et al., 2011). Rajalingham et al. (2018) additionally reported better behavior-CNN correspondence at the category but not at the individual exemplar level. Thus, comparing the representational structure at the category level, rather than at the exemplar level, should have increased our chance of finding a close brain-CNN correspondence. Yet despite the overall brain and CNN correlations for object categories being much higher here than in previous studies for individual objects (Khaligh-Razavi & Kriegeskorte, 2014; Cichy et al., 2016), CNNs failed to fully capture the representational structure of real-world objects in the human brain and performed even worse for artificial objects. Object category information is shown to be better represented by higher than lower visual regions (e.g., Hong et al., 2016). Our use of object category was thus not optimal for finding a close brain-CNN correspondence at lower levels of visual processing. Yet we found better brain-CNN correspondence at lower than higher levels of visual processing for real-world object categories. This suggests that information that defines the different real-world object categories is present at lower levels of visual processing and is captured by both lower visual regions and lower CNN layers. This is not surprising as many categories may be differentiated based on low-level features even with a viewpoint change, such as curvature and the presence of unique features (e.g., the large round outline of a face/head, the protrusion of the limbs in animals) (Rice et al., 2014)."

In terms of concrete changes I think should still be made: I think the authors should clarify whether the natural images were varied in orientation as the artificial ones were. If the natural images were not varied, then that difference should be clearly noted as a caveat to the difference in DNN performance in early visual areas. And I think the summary figure (Figure 7) would be better incorporating mean performance rather than whether DNNs captured 100% of the variance.

As mentioned above, we have now further clarified in the manuscript that the exemplars used for the natural images indeed varied in their appearances in terms of identity,

viewpoint/orientation and pose (for the animals). We have also included the full set of images used in Supplementary Figures 1 and 2 to add further clarifications. As the reviewer recommended, we have revised the summary figure (Figure 8) to show the mean performance across the different experiments rather than the success rate. We thank the reviewer for all the insightful critiques and suggestions which have helped us to further clarify and improve our results and arguments.

REVIEWERS' COMMENTS

Reviewer #1 (Remarks to the Author):

The authors have addressed my most recent set of comments, and I am satisfied with the manuscript.

Note that there may be minor typos in the paper; for example:

far part -> far apart

to a great extend -> to a great extent

Reviewer #2 (Remarks to the Author):

I think the manuscript is improved, and I recommend publication. The authors have added nuance and still more interesting data - the updates to the MDS plots and Figure 7 are nice. They have argued their case forcefully, and while I still disagree with a few of their points, I do not find my disagreements to be grounds to reject the paper.

To the authors: congratulations on a nice piece of work.

We thank the reviewers again for their time and effort and their comments on the manuscript. Below, we have addressed in detail each comment raised by the reviewers in the 3rd round of the review and have revised the manuscript accordingly. For easy referencing, we have included the reviewers' original comments in the italicized text below. In the revised manuscript, we have used track change to mark all the changes that we have made to the revised manuscript.

Reviewer #1 (Remarks to the Author):

The authors have addressed my most recent set of comments, and I am satisfied with the manuscript.

*Note that there may be minor typos in the paper; for example:
far part -> far apart
to a great extend -> to a great extent*

We apologize for these typos. We have done multiple additional careful read of the manuscript and we hope that we have caught all the remaining typos.

Reviewer #2 (Remarks to the Author):

I think the manuscript is improved, and I recommend publication. The authors have added nuance and still more interesting data - the updates to the MDS plots and Figure 7 are nice. They have argued their case forcefully, and while I still disagree with a few of their points, I do not find my disagreements to be grounds to reject the paper.

To the authors: congratulations on a nice piece of work.

We thank the reviewer for his comments.